# Tumor-anchored deep feature random forests for out-of-distribution detection in lung cancer segmentation

**Aneesh Rangnekar**                                                    *rangnea@mskcc.org*
*Department of Medical Physics*
*Memorial Sloan Kettering Cancer Center*

**Harini Veeraraghavan**                                                *veerarah@mskcc.org*
*Department of Medical Physics*
*Memorial Sloan Kettering Cancer Center*

**Reviewed on OpenReview:** *https://openreview.net/forum?id=XmjYlBxFxn*

## Abstract

Accurate segmentation of lung tumors from 3D computed tomography (CT) scans is essential for automated treatment planning and response assessment. Despite self-supervised pretraining on numerous datasets, state-of-the-art transformer backbones remain susceptible to out-of-distribution (OOD) inputs, often producing confidently incorrect segmentations with potential risk in clinical deployment. Hence, we introduce RF-Deep, a lightweight post-hoc random forests-based framework that uses deep features trained with limited outlier exposure, requiring as few as 40 labeled scans (20 in-distribution and 20 OOD), to improve scan-level OOD detection. RF-Deep repurposes the hierarchical features from the pretrained-then-finetuned segmentation backbones, aggregating features from multiple regions-of-interest anchored to predicted tumor regions to capture OOD likelihood.

We evaluated RF-Deep on 2,232 CT volumes spanning near-OOD (pulmonary embolism, COVID-19 negative) and far-OOD (kidney cancer, healthy pancreas) datasets. RF-Deep achieved AUROC > 93 on the challenging near-OOD datasets, where it outperformed the next best method by 4–7 percentage points, and produced near-perfect detection (AUROC > 99) on far-OOD datasets. The approach also showed transferability to two blinded validation datasets under the ensemble configuration (COVID-19 positive and breast cancer; AUROC > 94). RF-Deep maintained consistent performance across backbones of different depths and pretraining strategies, demonstrating applicability of post-hoc detectors as a safety filter for clinical deployment of tumor segmentation pipelines.

## 1 Introduction

Deep learning (DL)-based medical image analysis is now applicable throughout the continuum starting from diagnosis, treatment planning, and response monitoring at follow-up to enhance care of patients with cancer. For instance, hybrid transformer-convolutional networks have been shown to provide highly accurate tumor and healthy tissue segmentations from radiographic modalities (Jiang et al., 2022; Willemink et al., 2022; Nguyen et al., 2023; Yan et al., 2023; Qayyum et al., 2023; Gu et al., 2025). However, these methods are typically developed on narrowly-scoped datasets, and accuracy on in-distribution (ID) data does not translate to out-of-distribution (OOD) scenarios including different imaging acquisitions (Roschewitz et al., 2023; Koch et al., 2024), concept drifts (Huang et al., 2022; Sahiner et al., 2023; Gomez et al., 2025), and label shifts (Nichyporuk et al., 2022; Lempart et al., 2023; Godau et al., 2025). Importantly, DL models produce confidently inaccurate predictions (Hendrycks et al., 2019; Quiñonero-Candela et al., 2022; Prabhu et al., 2022; Banerjee et al., 2023), preventing clinicians from reliably using model uncertainties for decision making (Otles et al., 2021) and risking clinical expert deskilling through over-reliance on AI (Budzyń et al., 2025).

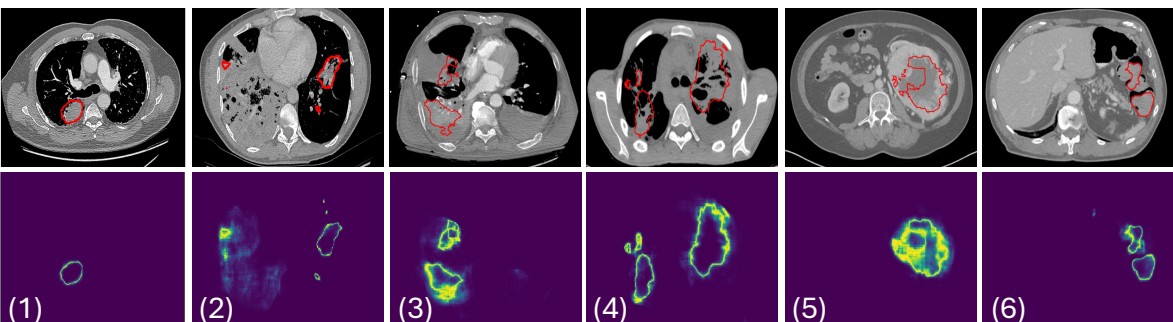

Figure 1: Uncertainty maps for in-distribution (ID) and out-of-distribution (OOD) scans. (a) ID lung cancer scans (1–2) show concentrated boundary uncertainty; OOD scans (3–6: pulmonary embolism, COVID-19-negative, kidney cancer, healthy abdomen) exhibit concentrated, diffused, or misaligned patterns.

Common deployment failures that motivate our work include Picture Archiving and Communication System (PACS) routing errors directing non-thoracic scans to lung-specific analysis pipelines as well as inadvertent introduction of out-of-scope thoracic diseases (e.g. pulmonary embolism, COVID-19 inflammation) mimicking tumor appearances. Segmentation accuracy metrics like the Dice similarity coefficient and Hausdorff distances do not reflect well-calibrated uncertainty estimates, even on ID datasets (Ren et al., 2024; Rangnekar et al., 2025). Whereas uncertainty visualizations can enhance clinical delineations (Goddard et al., 2012; Biase et al., 2024), such measures are also unreliable in OOD scenarios, as shown in Figure 1. Existing OOD detection methods are often too sensitive, flagging any deviation from training data (Roschewitz et al., 2023; Godau et al., 2025), limiting their use for practical purposes.

Our work overcomes the limitations of prior approaches by building on pretrained models to ensure sufficient generalization to common clinical imaging variations while enhancing robustness to far-OOD (disease in different anatomical sites) and near-OOD (different disease in same anatomic site) use cases. We propose a tumor-anchored deep feature extraction framework for post-hoc OOD detection. Unlike prior logit-based or distance methods that operate on entire images or require architectural modifications, we extract hierarchical encoder features from multiple regions of interest anchored to predicted tumor segmentations. This enables task-relevant detection that distinguishes subtle pathological differences within the same anatomical site while scaling to images with varying fields-of-view. Our contributions are:

- A post-hoc OOD detection approach using random forests with deep features (RF-Deep), trained with limited outlier exposure requiring as few as 40 labeled scans (20 ID + 20 OOD). Rather than aiming to detect arbitrary distribution shifts, RF-Deep is designed as a targeted safety filter to improve rejection of clinically relevant confounding cases, supporting more risk-reduced deployment of segmentation models.

- Improved robustness to common imaging variations through hierarchical backbone features from segmentation models pretrained using self-supervised learning (SSL) and subsequently fine-tuned for the downstream task. We empirically evaluate robustness across variations in scanner manufacturer, contrast usage, and reconstruction kernel.

- Evaluation on 2,232 3D CT scans from seven datasets for lung tumor segmentation, with analysis of SSL pretraining methods, model depths, and classifier designs on OOD detection performance.

- Mechanistic analysis of RF-Deep performance relative to radiomics-based and logit-based approaches using unsupervised clustering, SHAP-based feature importance, and spatial uncertainty analysis.

- External validation on two previously unseen OOD datasets (COVID-19-positive chest CT and breast cancer CT), providing encouraging evidence of transferability beyond the specific exposure distributions seen during training.

## 2 Related Works

Approaches for OOD detection fall into two classes: learner-based methods that extract distribution-robust features through training strategies, and filter-based methods that explicitly flag OOD scans at test time.

### 2.1 Learner-based methods

Learner-based methods extract robust feature representations from available training data so models generalize across domains without performance degradation. Domain-invariant features can be obtained via (a) specialized architectures (Saha et al., 2021; Simeth et al., 2023), (b) data augmentation in image and feature space (Jung et al., 2019; Chen et al., 2022; Vaish et al., 2025), (c) regularized training with multi-task objectives (Panfilov et al., 2019; Wen et al., 2024), and (d) SSL pretraining on large unlabeled examples (Hatamizadeh et al., 2022; Haghighi et al., 2022; Jiang et al., 2024; Du et al., 2024).

For tumor segmentation, architectures combining attention mechanisms and residual connections have achieved robust generalization across MR acquisitions (Saha et al., 2021; Simeth et al., 2023). Gu et al. (2021) trained separate domain-specific basis functions combined via group convolutions for prostate gland segmentation. Alternatively, data augmentation (Safdari et al., 2025) and regularization (Jung et al., 2019; Vaish et al., 2025) have been applied to standard architectures like U-Net (Chen et al., 2022; Wen et al., 2024) to further enhance robustness to distribution shifts (Boone et al., 2023).

SSL pretraining takes a different approach, exploiting unlabeled data through extensive data augmentations (Fedorov et al., 2021) or exposure to diverse imaging variations (Hatamizadeh et al., 2022; Jiang et al., 2024; Gomez et al., 2025). Azizi et al. (2023) proposed successive refinement, converting OOD data to ID through iterative adaptation, although unlike SSL, this requires labeled data for each new domain.

### 2.2 Filter-based OOD methods

Filter-based methods reject OOD scans at test time, ensuring models operate only on ID-like data where predictions are reliable (Berger et al., 2021). These methods implement either separate OOD detectors or task-integrated detection, and have been applied to handle imaging acquisition differences (Tardy et al., 2019; Gao & Wu, 2020; Nandy et al., 2021). Most filter-based methods focused on 2D classification, with approaches ranging from Mahalanobis distances (Tardy et al., 2019) to dedicated OOD networks (Nandy et al., 2021) to instance-level retraining that measured response variations on ID test sets (Gao & Wu, 2020). A different approach involves using a single model to classify tasks on inlier data, while exposing the same model to outlier examples to detect OOD during inference (Roy et al., 2022; Araujo et al., 2023). This approach is practical when the OOD types are known from domain knowledge (e.g., optical tomography versus retinal fundus images) (Araujo et al., 2023), or when outliers such as infrequent disease conditions are identifiable from the training data (Roy et al., 2022).

For segmentation, Mahalanobis distance on feature embeddings has been applied to COVID-19 lung lesion detection (González et al., 2022), but the high dimensionality of feature embeddings limits accuracy, particularly for distinguishing diseases within the same disease site (Berger et al., 2021; Roy et al., 2022). Dimensionality reduction via principal component analysis (PCA) (Woodland et al., 2024) or VQ-GAN (Pinaya et al., 2022; Graham et al., 2023) has been used to reduce the latent space and sharpen the separation of ID from OOD feature distributions, but introduces inductive bias (Nalisnick et al., 2019) and requires additional training. Logit-based scores (Hendrycks & Gimpel, 2017; Hendrycks et al., 2022) and temperature scaling (Karimi & Gholipour, 2022) offer computational simplicity but inherit task-model biases (Mehrtash et al., 2020). Architectural modifications (Yuan et al., 2023; Larrazabal et al., 2023) and generative diffusion methods (Nguyen et al., 2026) have improved OOD detection but increase network parameters and training complexity. Radiomic feature distances have shown utility for dataset-level OOD detection (Vasiliuk et al., 2023; Konz et al., 2024) but struggle with scan-level detections, as they cannot simultaneously capture subtle data variations while providing robust performance to common imaging variations.

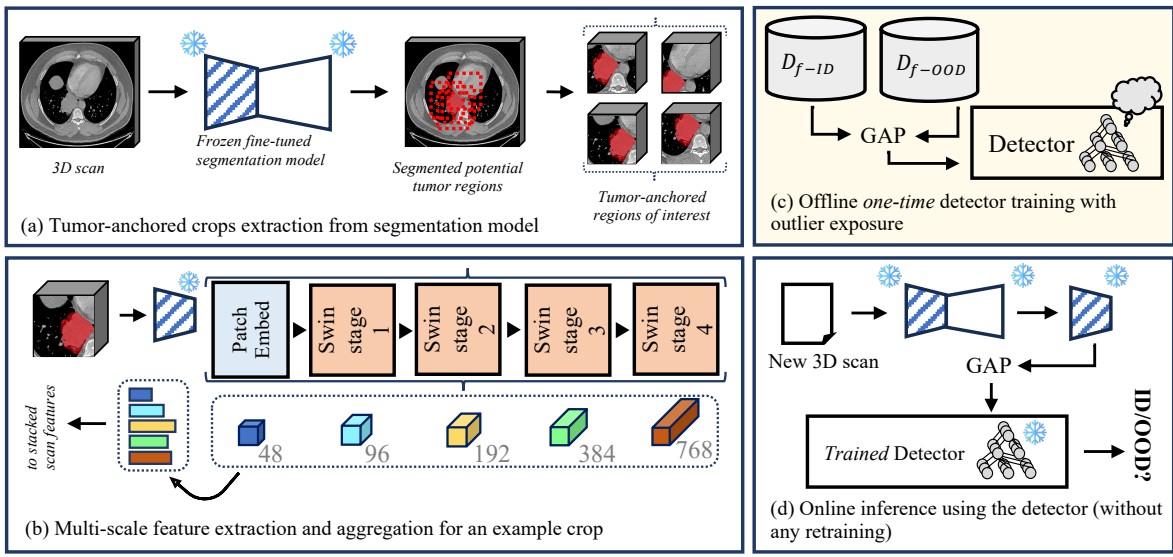

Figure 2: RF-Deep workflow for scan-level OOD detection. (a–c) Feature extraction from the frozen ❄ segmentation model and random forest training in an outlier exposure manner; (d) scan-level ID/OOD inference using the trained detector.

## 3 Framework overview and task definition

To address these limitations, our goal was to develop a highly accurate scan-level OOD detection approach, which does not require large numbers of outlier samples in practice, with applicability to images with different acquisitions and fields of view. Let $\mathcal{D}_{\text{in}}$ denote the distribution of lung cancer CT scans, closely matching the training dataset used to create the lung tumor segmentation model, and let $\mathcal{D}_{\text{out}}$ represent the scans that differ in pathology and anatomic sites. Given a new scan $x$, the objective is to determine whether it belongs to $\mathcal{D}_{\text{in}}$ or $\mathcal{D}_{\text{out}}$ at the scan level using a scoring mechanism provided by a random forest (RF) classifier.

Our approach combines the strengths of learner-based and filter-based OOD detection methods. Specifically, we leverage the backbone of a segmentation model pretrained via self-supervised learning to extract imaging features that are largely domain-agnostic, and pair these features with a lightweight random forest classifier to reject distributionally dissimilar scans. The RF-based OOD classifier incorporates known outliers during training, an outlier exposure strategy shown to be effective in prior work (Hendrycks et al., 2019; Thulasidasan et al., 2021; Guha Roy et al., 2022). We adopt outlier exposure for three key reasons. First, in clinical deployment, common OOD failure modes (e.g., incorrect body region or scanner-related differences) are often predictable based on domain knowledge. Second, purely unsupervised methods (such as Mahalanobis distance–based approaches) frequently struggle with near-OOD scans (Table 1), where diseases share anatomical context with the in-distribution data. Third, our method requires only a small number of outlier examples (as few as 20 scans) to achieve strong performance (Section 5.5), making outlier curation feasible in practice. A core assumption of this approach is that exposure to even a small, diverse set of OOD examples provides sufficient signal for the RF to learn ID-vs-OOD decision boundaries that generalize beyond the specific outliers seen during training; an assumption supported empirically by the blinded validation results (Section 5.2).

Importantly, we evaluate generalization to *unseen* OOD categories via external validation on two blind test datasets (Section 5.2) and leave-one-dataset-out experiments across deployment strategies (Section 3.3), providing evidence that RF-Deep may transfer beyond the specific exposure distributions encountered during training. Scan-level OOD prediction for segmentation involves thousands of voxel-level predictions, which, while offering rich spatial context, also introduces memory constraints in processing and aggregating information from their large 3D volumes. Our approach addresses this issue by focusing the OOD detection in regions of interest (ROI) anchored to the predicted tumor segmentation, and then averaging the metric

scores from those regions, which also allowed the approach to scale to images with varying fields of view. Using regions anchored near the predicted tumors also allows the model to focus on differentiating features in relevant parts of the image, as opposed to distinguishing entire images, enabling distinction of diseases occurring in the same disease site.

A key design consideration is RF-Deep's dependence on the segmentation model's predictions. We observed that on OOD scans, the segmentation model typically produced confidently incorrect tumor-like regions, rather than failing to produce any output. These predictions served as anchors for ROI extraction, and the extracted features from these regions are precisely what RF-Deep uses to identify the scan as OOD. In rare cases where the model produces no foreground predictions, the absence itself can serve as an implicit OOD signal (by assigning the maximum OOD score of 1.0 and flagging the scan for further review), as ID scans with confirmed tumors would be expected to yield predictions. Across our evaluation datasets, the segmentation model produced at least one connected component of most OOD scans, confirming that the tumor-anchored approach remains applicable even under distribution shift.

## 3.1 Segmentation model

The segmentation framework employs a hybrid transformer-convolution architecture that combines the transformer backbone as the encoder for global contextual modeling with a convolutional decoder for local spatial precision. The backbone is a hierarchical Swin Transformer (Liu et al., 2021) with depth configuration $2-2-12-2$ across four successive stages, with a patch size of $2 \times 2 \times 2$ voxels and a window size of $4 \times 4 \times 4$ voxels. Its hierarchical design progressively aggregates features at multiple spatial scales to capture fine-grained anatomical details. Windowed self-attention within each stage reduces complexity while preserving details and global context, critical for processing large volumetric images of size $128 \times 128 \times 128$ voxels. The backbone was pretrained with the self-distilled masked image transformer (SMIT) framework to enhance robustness to imaging and patient variations (Jiang et al., 2022). SMIT combines masked image prediction (He et al., 2021; Xie et al., 2022) with self-distillation (Zhou et al., 2022); described in Appendix A. The decoder, a 3D U-Net-based (Ronneberger et al., 2015) convolutional structure, was randomly initialized and fine-tuned on the ID task-specific dataset. Following training, the entire model is frozen (❄) for subsequent segmentation and feature-based OOD detection.

## 3.2 Scan-level OOD detection

Our OOD detection pipeline consists of the following four steps (Figure 2):

- **Regions of interest extraction:** The task-specific segmentation model (Subsection 3.1) was used to generate tumor segmentations from the entire scan. Detected tumor regions were used to extract multiple 3D ROIs via random spatial offsets, ensuring the tumor appears within each ROI without precise centering (Figure 2a).

- **Feature extraction.** We repurposed the fine-tuned backbone encoder from the segmentation model to extract multi-scale features from the five encoder stages: the patch embedding layer (48 channels) and the four Swin Transformer stages (96, 192, 384, and 768 channels, respectively). Features were aggregated into a single scan-level descriptor using global average pooling (GAP) over tumor-anchored ROIs, producing a scan-level feature vector (Figure 2b). This tumor-conditioned aggregation emphasizes spatially relevant representations for downstream OOD discrimination.

- **Training the OOD detector.** Feature vectors were extracted from ID lung cancer datasets as well as a representative set of OOD scans from tumor-anchored regions as described above and used to train the RF classifier. This approach enabled our detector to learn subtle decision boundaries that distinguish shifts even between different diseases affecting the same anatomic region (Figure 2c).

- **Online inference.** Given a new test instance, the aforementioned pipeline extracts potential tumor regions and tumor-anchored ROIs, followed by the extraction of GAP features, which the RF classifier uses to generate scan-level OOD classification (Figure 2d).

### 3.3 Deployment strategies

RF-Deep supports two complementary deployment strategies. In the *dataset-specific* mode, a separate RF classifier was trained for each anticipated OOD failure mode (for example, pulmonary embolism), providing the highest detection accuracy when the OOD type is predictable from domain knowledge. This design is intended to maximize discrimination when the deployment shift closely resembles the outlier exposure distribution. We additionally used dataset-specific classifiers to test for scanner and protocol shortcuts by aggregating their average OOD detections across individual models.

In the *ensemble* mode, predictions from multiple dataset-specific classifiers are averaged at test time without knowledge of the test domain, providing robust detection of both seen and unseen OOD categories. This strategy is intended to improve robustness across heterogeneous cohorts by balancing specialization with generalization, and is therefore used when the OOD type is unknown. The ensemble configuration was also used to test for dataset-specific shortcuts, as consistent performance across held-out domains would indicate that detection is not driven by acquisition-level biases.

Finally, because the OOD type of an incoming scan is unknown at inference time, the ensemble configuration is the recommended default for real-world deployment; the dataset-specific mode is reserved for settings where the failure mode can be reliably anticipated.

### 3.4 Deployment considerations

To facilitate practical adoption and clarify the operational overhead of RF-Deep, we formalize the three core requirements for deployment:

1. **Frozen Segmentation Backbone:** The method is strictly post-hoc. It requires no access to the pretraining or fine-tuning data of the segmentation model and no modifications (retraining or fine-tuning) to the model's architecture or weights.

2. **Limited Labeled Exposure:** Training the random forest classifier requires only a small calibration set. As shown in our sensitivity analysis (Figure 10c), performance remains robust with as few as 40 total labeled scans (20 ID and 20 OOD).

3. **Default Hyperparameters:** RF-Deep achieves high OOD detection performance using default random forest hyperparameters, eliminating the need for intensive validation-set tuning or expert intervention during setup.

## 4 Experimental setup

### 4.1 Implementation details

All models were implemented in PyTorch (Paszke et al., 2019) and MONAI (Cardoso et al., 2022). Our method operates on full 3D CT volumes at full resolution, resampled to a uniform isotropic voxel spacing of $1 \text{ mm}^3$. All scans were clipped to the lung window (Hounsfield units (HU): $[-400, 400]$) and then intensity-normalized. During inference, the segmentation model processes the entire volume using sliding window inference ($128 \times 128 \times 128$ voxel patches with 50% Gaussian overlap) to generate tumor segmentations. For OOD detection, every detected tumor seeds multiple tumor-anchored 3D ROIs ($128 \times 128 \times 128$ voxels each) around the predicted tumor areas; the OOD scores from these regions are combined to generate a scan-level prediction.

**Segmentation.** Models were fine-tuned using cross-entropy and Dice loss with a batch size of 16 distributed across four NVIDIA GPUs, with an input size of $128 \times 128 \times 128$ voxels. We used AdamW optimizer (Loshchilov & Hutter, 2017) with a learning rate of $2 \times 10^{-4}$, linear warm-up (50 epochs), and cosine annealing over 1,000 epochs. Data augmentations included random flips, rotations, affine transformations, and illumination adjustments, consistent with Tang et al. (2022); Jiang et al. (2022).

**OOD Detection.** Random forest classifiers used 1,000 trees with a maximum depth of 20 with default *scikit-learn* hyperparameters. Performance remained stable across a broad range of hyperparameter settings, and this configuration was applied consistently across datasets without per-cohort tuning. We extracted deep feature representations from $n = 4$ tumor-anchored 3D ROIs per scan for all ID and OOD datasets. All ROIs were used individually during training to increase training samples, and probabilities from all ROIs were averaged at the scan level as an ensemble for inference. For MaxSoftmax, MaxLogit, and energy, scan-level scores were obtained by mean-aggregating tumor (i.e. positive class) predictions across all voxels using the corresponding metric.

## 4.2 Datasets

**Backbone pretraining.** The backbone encoder was pretrained using the self-distilled masked image transformer (SMIT) framework on 10,432 3D CT scans, encompassing public and institutional datasets, following Jiang et al. (2022); Tang et al. (2022); Jiang et al. (2024).

**Comparison to other pretraining strategies and architectures.** Additionally, we pretrained using SimMIM (Xie et al., 2022), iBOT (Zhou et al., 2022), and Swin UNETR (Tang et al., 2022) to assess the impact of pretraining strategy on both accuracy and OOD robustness. To analyze the effect of model depth, we pretrained a lite backbone configuration $(2 - 2 - 2 - 2)$ using SMIT. Finally, we evaluated the publicly available Swin UNETR pretrained checkpoint (marked as $^\dagger$) to assess the impact of pretraining data.

We do not consider interactive foundation models for segmentation, such as SAM (Kirillov et al., 2023) or its medical variants (e.g., MedSAM (Ma et al., 2024) and VISTA3D He et al. (2025)), as these methods are primarily designed for prompt-guided segmentation and require user-provided inputs (points or bounding boxes), which precludes fully automated deployment. Additionally, we exclude self-supervised 2D models such as DINOv2 (Oquab et al., 2023), which are pretrained on natural images and operate on individual slices, thereby losing critical inter-slice spatial context necessary for distinguishing 3D pathological structures in CT volumes (that is, relying on 2.5D processing).

**Fine-tuning.** We combined the pretrained backbone as the encoder with a randomly initialized U-Net decoder and fine-tuned using $N = 317$ 3D scans from the publicly available non-small cell lung cancer (NSCLC) Radiomics dataset (Aerts et al., 2015) with radiologist-provided tumor delineations. The original dataset contains 422 scans; 317 were selected based on radiologist assessment and verification of tumor boundary quality.

**Testing.** We used the NSCLC Radiogenomics dataset (Bakr et al., 2017) as the ID test set. From the original 211 scans, $N = 140$ scans from various scanners (GE, Siemens) were selected based on radiologist assessment and verification of tumor boundaries and image quality. The ID test set is completely separate from the training set, acquired from different patients and lung cancer stages, providing external validation. OOD evaluation used 1,916 3D CT scans from four public datasets: pulmonary embolism (RSNA PE, $N = 1,225$ selected from 3,114 scans, preserving the marginal distribution of labels in the full PE-positive dataset) (Colak et al., 2021), COVID-19 negative patients (MIDRC C19$^-$, $N = 120$, all used) (Tsai et al., 2021), kidney cancer (KiTS, $N = 489$, all used) (Heller et al., 2023), and healthy scans of the pancreas (PancreasCT, $N = 82$, all used) (Roth et al., 2015). The near-OOD datasets (RSNA PE, MIDRC C19$^-$) consist of thoracic (chest) CT scans involving non-cancerous diseases such as pulmonary embolism and inflammation, depicting appearances similar to lung lesions. The far-OOD datasets (KiTS, PancreasCT) include scans with larger field-of-view than training with healthy organs and non-lung cancers. Patient-level splits ensured zero overlap between training and evaluation; OOD counts were held constant across runs.

**Blinded validation.** To further evaluate generalization beyond seen OOD datasets, we additionally evaluated on two datasets that were never included in any OE training split: COVID-19-positive chest CT (MIDRC C19$^+$; $N = 110$) with active COVID-19 infection Tsai et al. (2020) and breast cancer CT ($N = 66$; institutional dataset, citation withheld for double-blind review) representing a near-OOD with shared thoracic anatomy but distinct tumor pathology from the ID lung cancer cohort (Figure 4). The datasets were treated as fully blind to assess generalization to novel thoracic diseases without dataset-specific adaptation.

### 4.3 Evaluation metrics

Segmentation accuracy on ID datasets was measured using the Dice similarity coefficient (DSC) and the 95th percentile Hausdorff distance (HD95). OOD detection accuracy was computed using the area under the receiver operating characteristic curve (AUROC) and false positive rate at 95% true positive rate (FPR at 95%TPR, or FPR95). AUROC quantifies the separability between ID and OOD scans (Davis & Goadrich, 2006), with higher values indicating better discrimination. FPR95 measures the proportion of OOD scans misclassified as ID when sensitivity is fixed at 95% (Liang et al., 2018), where lower values are better.

To ensure meaningful AUROC computation, the number of OOD scans was balanced with ID test scans ($N = 140$) by subsampling $N = 140$ OOD examples without replacement per run from larger datasets (RSNA PE and KiTS) and averaging the computed metrics. Splits ensured zero patient overlap with consistent OOD scan counts. This evaluation approach avoids biases from imbalanced evaluation sets while ensuring stable ID-OOD ratios, following best practices for robust OOD evaluation (Szyc et al., 2023; Humblot-Renaux et al., 2023).

For segmentation performance, we reported the mean and standard deviation across test scans. For OOD detection performance, we reported the point estimates with 95% bootstrap confidence intervals computed over 100 runs with matched random seeds to reflect sampling variability across different train-validation splits.

### 4.4 Comparative OOD detection methods

Baseline comparisons included commonly used OOD detection methods, namely, MaxSoftmax (Hendrycks & Gimpel, 2017), MaxLogit (Hendrycks et al., 2022), and energy (Liu et al., 2020), performed by computing scan-level scores by mean-aggregating the tumor (positive class) predictions across all voxels (see Appendix B).

In addition, a RF classifier was created using handcrafted radiomics features ($N = 293$) called RF-Radiomics, computed within the segmented tumor regions following the Image Biomarker Standardization Initiative (IBSI) guidelines (Zwanenburg et al., 2020) with open-source software PyCERR (Apte et al., 2018). Critically, RF-Radiomics uses *identical* outlier exposure as RF-Deep (same ID and OOD training splits), isolating the effect of feature representation: any performance difference between RF-Deep and RF-Radiomics is attributable to deep features versus handcrafted features, not to differential access to OOD supervision. Details of individual radiomic features are in Table 9. The high correlation among radiomic features was addressed by employing recursive feature elimination to select a more compact and informative set of radiomic features. Of note, recursive feature elimination was not used for training the RF-Deep classifier.

Finally, the Mahalanobis distance-based OOD detector (MD-Deep) (Lee et al., 2018) was computed with respect to deep features of the ID data, modeled as a single Gaussian in feature space using Ledoit-Wolf covariance estimation. Hence, MD-Deep computes OOD scores as the distance of each scan with respect to the fitted ID distribution. MD-Deep uses identical backbone features as RF-Deep but operates without outlier exposure, isolating the contribution of OE and the RF classifier's learned decision boundaries. Comparing RF-Deep to MD-Deep thus reveals the benefit of outlier exposure and non-linear classification over purely unsupervised density estimation on the same feature space.

We additionally boosted MD-Deep with post-hoc feature-space methods (ReAct (Sun et al., 2021), ASH (Djurisic et al., 2022), and X-Mahalanobis (Wei et al., 2025)), but these methods did not improve over standard MD-Deep on our spatially-pooled encoder (Appendix C). We excluded ODIN (Liang et al., 2018) because it relies on temperature scaling and input perturbations calibrated on fine-tuning data, parameters that are often unavailable when deploying models to new out-of-distribution clinical datasets. Finally, we also excluded KNN-based approaches (Sun et al., 2022) and ViM (Wang et al., 2022) as they are similarly unsupervised, relying solely on ID feature density without outlier exposure that our results show is ineffective for near-OOD detection (Table 1).

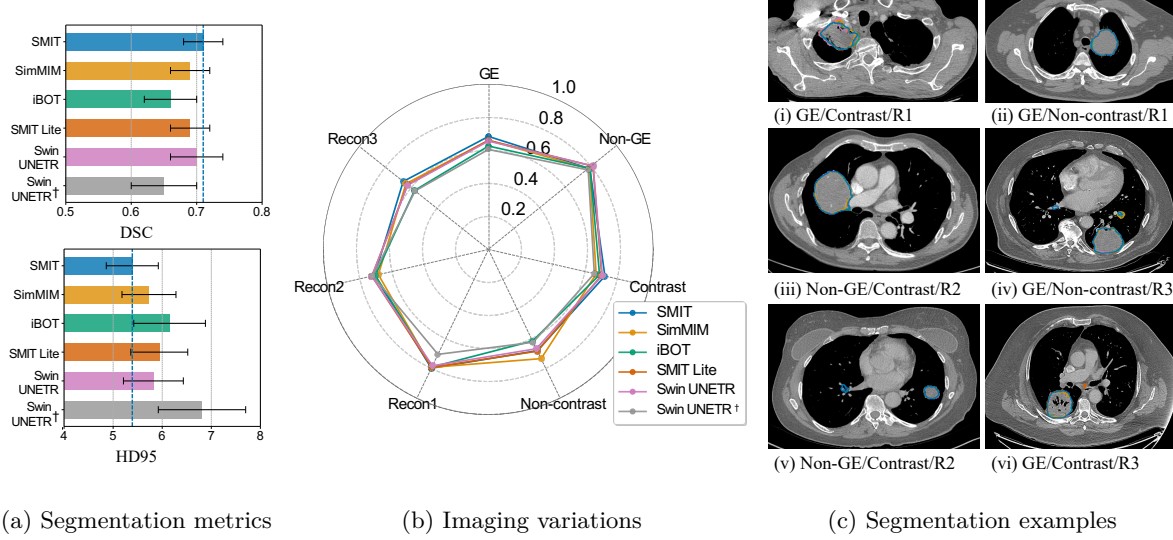

(a) Segmentation metrics      (b) Imaging variations      (c) Segmentation examples

Figure 3: Pretraining strategies performance and robustness evaluation on ID test set. (a) Segmentation metrics (DSC and HD95). (b) Performance across imaging variations. (c) Representative segmentations; $R$ denotes reconstruction kernel.

## 5 Results

### 5.1 Segmentation performance on ID dataset

Segmentation performance across pretraining strategies is shown in Figure 3a for the ID testing dataset. All pretrained methods were similarly accurate, with iBOT producing the lowest accuracy, followed by the publicly available Swin UNETR[†] checkpoint.

Next, we performed a disaggregated analysis of model performance under common imaging variations including scanner manufacturer (GE versus non-GE), contrast versus non-contrast enhanced CT, and reconstruction kernels. Reconstruction kernels were categorized as Recon 1 (GE 'standard' and 'bone', Siemens with < B40), Recon 2 (GE 'Bone Plus', Siemens with ≥ B40 and < B50), and Recon 3 (GE 'Lung', Siemens ≥ B50). Segmentation model fine-tuning was performed on the public dataset acquired on Siemens scanners with contrast.

From Figure 3b, all models were similarly accurate across the analyzed variations, with larger gaps in accuracy noted for Swin UNETR[†] (Recon 1, non-contrast CT) and iBOT (Recon 3, non-contrast). SMIT and SMIT Lite strategies were similarly accurate, indicating that a smaller model is sufficient for robust tumor segmentation. On the other hand, choice of pretraining data impacted accuracy, as seen by the performance difference between the Swin UNETR trained with the 10K dataset and the public checkpoint Swin UNETR. Detailed accuracies using DSC and HD95 metrics for the analyzed methods are shown in Figure 12. Representative segmentation examples are shown in Figure 3c.

Overall, the SMIT-pretrained model was the most accurate, followed by the Swin UNETR method, and was therefore selected for benchmarking RF-Deep throughout this paper.

### 5.2 Out-of-distribution detection performance

RF-Deep is the most accurate method for both near-OOD and far-OOD datasets (Table 1; using the SMIT-pretrained backbone). On the near-OOD datasets (RSNA PE and MIDRC C19$^-$), it achieved AUROCs of 95.80 and 93.30, improving AUROC by 4–7 percentage points over the next best method per dataset, and

Table 1: OOD detection performance comparing various methods on the segmentation model fine-tuned using the SMIT-pretrained backbone. AUROC ($\uparrow$) and FPR95 ($\downarrow$) are reported with 95% bootstrap confidence intervals over 100 matched-seed runs.

| Method | RSNA PE | | MIDRC C19$^-$ | | KiTS | | PancreasCT | |
|---|---|---|---|---|---|---|---|---|
| | AUROC | FPR95 (%) | AUROC | FPR95 (%) | AUROC | FPR95 (%) | AUROC | FPR95 (%) |
| MaxSoftmax | 88.61 | 38.37 | 86.57 | 49.27 | 95.67 | 23.26 | 94.61 | 34.27 |
| | (84.62–92.36) | (31.78–49.24) | (81.79–90.14) | (39.53–59.78) | (93.41–98.06) | (11.47–34.19) | (92.61–97.49) | (24.80–49.30) |
| MaxLogit | 88.77 | 40.01 | 89.31 | 41.59 | 95.89 | 18.05 | 93.53 | 30.87 |
| | (85.09–92.43) | (24.80–52.52) | (85.05–93.06) | (29.84–53.96) | (93.58–98.44) | (10.41–27.37) | (90.17–96.59) | (16.17–56.87) |
| Energy | 88.59 | 39.86 | 89.30 | 43.04 | 95.80 | 17.47 | 93.52 | 29.96 |
| | (84.85–92.24) | (24.80–52.55) | (84.98–93.04) | (29.84–55.43) | (93.41–98.44) | (10.41–26.65) | (90.10–96.53) | (15.83–56.87) |
| RF-Radiomics | 88.70 | 41.10 | 91.30 | 30.00 | 96.20 | 11.80 | 96.10 | 17.20 |
| | (81.50–93.00) | (25.20–57.80) | (87.10–94.40) | (17.40–45.60) | (93.00–98.30) | (5.80–21.90) | (89.10–99.20) | (4.30–41.90) |
| MD-Deep | 87.30 | 45.10 | 84.50 | 58.50 | 99.20 | 3.10 | 99.20 | 2.80 |
| | (83.90–91.10) | (30.60–60.30) | (79.40–89.00) | (38.20–77.60) | (97.90–99.80) | (0.00–9.20) | (98.20–100.00) | (0.00–7.70) |
| **RF-Deep** | **95.80** | **15.20** | **93.30** | **25.50** | **100.00** | **0.10** | **100.00** | **0.00** |
| | **(92.60–98.50)** | **(7.10–23.50)** | **(89.90–96.00)** | **(16.80–36.20)** | **(99.90–100.00)** | **(0.00–1.00)** | **(100.00–100.00)** | **(0.00–0.00)** |

reducing FPR95 by 18–25 absolute points, with non-overlapping confidence intervals, indicating consistent superiority.

On the far-OOD datasets (KiTS and PancreasCT), RF-Deep achieved near-perfect detection with near-zero false alarms for anatomically distant diseases (and disease sites). These results are expected given the substantial anatomical differences and serve as a "sanity check" validation that the detector does not fail on straightforward scans. Notably, the gap between near- and far-OOD performance is smaller for RF-Deep than for competing approaches, indicating improved robustness to subtle distribution shifts.

Compared to RF-Radiomics (same classifier and outlier exposure, but different features), RF-Deep demonstrated that deep features better capture subtle differences, particularly in near-OOD datasets (analyzed further in Sections 5.3.1 and 5.3.2). Compared to MD-Deep (same features, no outlier exposure), the results highlighted the importance of outlier exposure and non-linear decision boundaries. Additional evaluation to assess whether feature reduction improves MD-Deep performance (Figure 13) showed no gains over the standard formulation. In contrast to all logit-based methods, RF-Deep showed that the encoder features provided more discriminative signals than output-layer uncertainties. For instance, MaxLogit exhibited performance gaps of 7–8% between near-OOD and far-OOD datasets, more than twice the gap observed for RF-Deep (3.5–6.7%). We analyzed the underlying causes of these performance differences in Section 5.3.3.

### 5.2.1   Transferability to blinded validation datasets

To evaluate generalization under the recommended deployment configuration, we report results exclusively under the ensemble detector, as this represents the operationally appropriate setting when the OOD type is unknown at inference time. RF-Deep achieved an AUROC of 94.71 on MIDRC C19$^+$ and 97.32 on breast cancer CT under the ensemble configuration, outperforming competing approaches (Table 2). Notably, performance on the unseen breast cancer CT exceeded that on the seen near-OOD datasets (Table 6), suggesting that RF-Deep's learned decision boundaries may transfer effectively to related thoracic distribution shifts even without direct exposure during training.

These results were consistent with the primary results in Table 1. Unsupervised MD-Deep struggled on challenging near-OOD scans, while radiomics features (RF-Radiomics) underperformed relative to the deep features. RF-Deep maintained superior performance, indicating that combining hierarchical features with limited outlier exposure improved robustness to previously unseen distribution shifts. Logit-based methods achieved moderate performance on both datasets but remained 3–5 AUROC percentage points below RF-Deep on MIDRC C19$^+$. As these datasets served to validate generalization rather than as a full benchmark, we reported comparisons on the SMIT backbone only; backbone-level analyses are reported on the primary four OOD datasets where controlled ablations are available.

Table 2: OOD detection on blinded validation datasets completely unseen during OE training.

| Method | MIDRC C19$^+$ | | Breast cancer CT | |
|---|---|---|---|---|
| | AUROC | FPR95 (%) | AUROC | FPR95 (%) |
| MaxSoftmax | 89.37 | 36.32 | 91.86 | 34.62 |
| | (85.12–92.71) | (25.36–52.55) | (87.47–95.45) | (24.98–50.83) |
| MaxLogit | 89.89 | 32.81 | 94.27 | 23.15 |
| | (85.66–93.77) | (22.08–52.21) | (91.13–97.11) | (15.94–30.82) |
| Energy | 89.54 | 32.62 | 94.34 | 22.86 |
| | (85.09–93.62) | (22.08–52.17) | (91.36–97.16) | (15.56–30.82) |
| RF-Radiomics | 88.43 | 35.65 | 96.90 | 7.99 |
| | (82.39–93.89) | (24.73–49.05) | (91.49–99.17) | (3.37–18.50) |
| MD-Deep | 71.15 | 95.90 | 96.40 | 9.81 |
| | (61.83–80.49) | (83.14–100.00) | (93.41–99.24) | (2.04–19.39) |
| **RF-Deep** | **94.71** | **25.91** | **97.32** | **9.78** |
| | **(91.62–96.78)** | **(16.32–37.34)** | **(94.32–99.60)** | **(2.01–22.08)** |

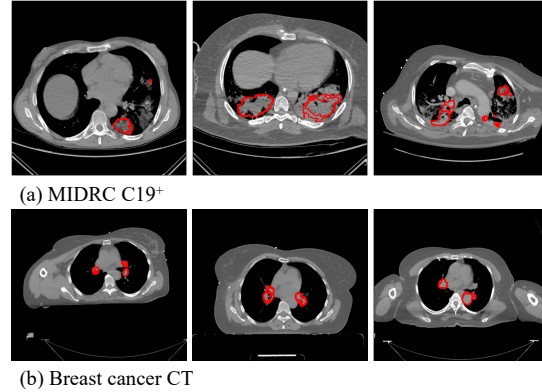

(a) MIDRC C19$^+$

(b) Breast cancer CT

Figure 4: Representative slices from blinded validation datasets unseen during OE training. Reported with 95% bootstrap confidence intervals over 100 matched-seed runs.

### 5.2.2 Impact of model pretraining on OOD detection

Next, we examined the impact of backbone pretraining and architecture on OOD detection performance. RF-Deep remained the most accurate method across all backbone architectures (Table 12). Accuracy differences between SMIT and SMIT Lite models on near-OOD datasets were modest, with SMIT Lite showing slightly better performance on RSNA PE but marginally lower on MIDRC C19$^-$. Both Swin UNETR variants exhibited 4–7 percentage points lower AUROC on the near-OOD datasets compared to SMIT-based models, reaffirming that pretraining strategy and encoder architecture substantially influence OOD detection performance within the RF-Deep framework.

### 5.2.3 Impact of detection characteristics on OOD scoring

To assess whether OOD detection was influenced by properties of the predicted foreground, we analyzed both (i) the relationship between predicted lesion volume and OOD probability, and (ii) the behavior of scans with no retained predictions.

From Figure 5, a moderate positive correlation was observed for RSNA PE ($\rho$=0.48 and 0.39 for dataset-specific and ensemble, both $p < 0.001$), indicating that larger detections were assigned higher OOD probability and that smaller detections yielded less distinctive encoder representations relative to the ID distribution, reducing their separability. No such relationship was found for MIDRC C19$^-$ under the ensemble detector ($\rho$=0.16, $p = 0.214$), suggesting that detection in that cohort is not primarily size-driven. Additionally, the same volume-dependent pattern was observed in both blinded validation datasets ($\rho = 0.36$ and 0.56, both $p < 0.001$), with results identical across detectors, providing encouraging evidence that the relationship generalizes beyond the OE training distribution.

To further probe this effect, we stratified scans by predicted foreground volume based on the ID statistics and evaluated detection performance on the smallest-volume bin (Q1: 0–4.32 cc). For far-OOD datasets (KiTS, PancreasCT), detection remained robust, with no small-volume scans misclassified (0/33 and 0/27, respectively) and high mean OOD probabilities (0.82–0.93), indicating that anatomical mismatch dominates detection irrespective of lesion size. In contrast, near-OOD datasets showed a marked degradation for small lesions, particularly under the ensemble detector (RSNA-PE: 69%; MIDRC-C19$^-$: 54%; mean probability $\approx 0.45$), with predictions concentrated near the decision boundary rather than being confidently misclassified. The dataset-specific detector remained comparatively robust (RSNA-PE: 6%; MIDRC-C19$^-$: 17%; mean probability $\approx 0.67$), reflecting its more-informed decision due to the access to specific target-domain characteristics. Notably, the same pattern extended to the blinded validation datasets, where both detectors behaved identically on small lesions (MIDRC-C19$^+$: 64%; breast cancer CT: 9/13, 69%), indicating that the

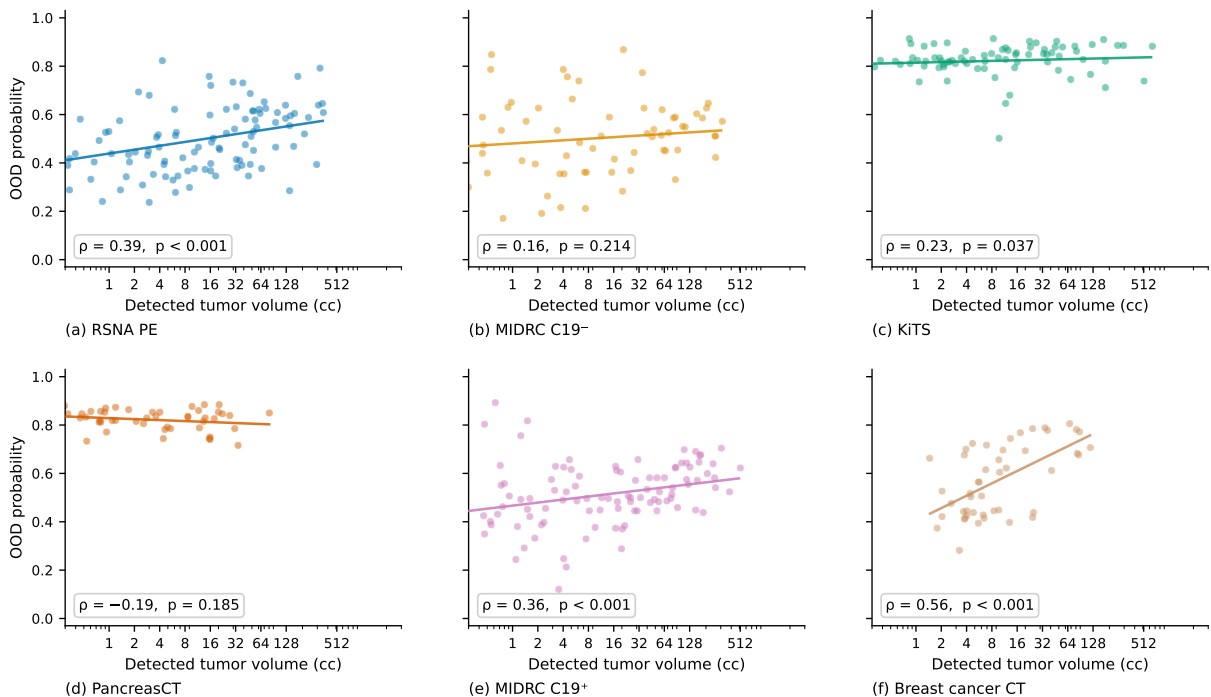

Figure 5: OOD probability as a function of predicted foreground volume. Results from one representative split (of 100), combining all datasets in a single visualization, are shown for brevity. Each point represents one scan; the line shows the least-squares fit on log-transformed volume. Spearman $\rho$ and associated $p$-value are shown per panel. Panels (a)–(d) correspond to the four standard OOD datasets; (e)–(f) are blinded validation datasets not included in OE training.

difficulty arises from detecting small, ambiguous near-OOD lesions without prior exposure rather than from a detector-specific failure.

In addition, we analyzed the scenarios where the segmentation model did not predict any tumors in OOD datasets. Of note, this is the favorable scenario as it implies the segmentation model has captured the data distribution properly. Across OOD datasets, no-prediction rates varied substantially: RSNA PE (10.1%), MIDRC C19$^-$ (32.3%), KiTS (32.2%), and PancreasCT (34.2%). In contrast, fewer than 3 out of 98 ID scans produced zero foreground predictions. We evaluated a simple enhanced pipeline, assigning an OOD score of 1.0 to all scans without any predictions. This pipeline matched or improved AUROC on near-OOD datasets while only minimally affecting far-OOD performance (Table 3). It introduced only 2.73 mean ID false positives per run ($\sim$2.8%), indicating limited impact on in-distribution specificity. Notably, higher no-prediction rates in far-OOD datasets (e.g., KiTS and PancreasCT $\sim$32–34%) suggest that the absence of predictions acts as a complementary and anatomically meaningful OOD signal, rather than merely reflecting detector failure.

## 5.3 Why RF-Deep outperforms alternatives?

To better understand the performance differences observed above, we analyzed the empirical factors underlying RF-Deep's improved performance.

### 5.3.1 RF-Deep extracts features that are better separated than RF-Radiomics

Figure 6 visualizes the t-SNE embeddings (perplexity=30, 1,000 iterations; consistent across random seeds) of deep features (panel a) and radiomics features (panel b) for the ID and OOD datasets collectively. Deep features exhibit better cluster separation between the ID and OOD datasets, with minimal overlap between

Table 3: RF-Deep with standard and enhanced pipelines (score = 1.0 on absent predictions). 95% percentile CI across 100 matched-seed runs.

| Method | RSNA PE | MIDRC C19$^-$ | KiTS | PancreasCT |
|---|---|---|---|---|
| *Dataset-Specific* | | | | |
| *AUROC* | | | | |
| Standard | 95.80 (92.60–98.50) | 93.30 (89.90–96.00) | 100.00 (99.90–100.00) | 100.00 (100.00–100.00) |
| Enhanced | 96.14 (92.80–98.29) | 95.43 (92.53–97.34) | 99.42 (99.07–100.00) | 99.44 (99.12–100.00) |
| *FPR95 (%)* | | | | |
| Standard | 15.20 (7.10–23.50) | 25.50 (16.80–36.20) | 0.10 (0.00–1.00) | 0.00 (0.00–0.00) |
| Enhanced | 15.43 (6.61–24.49) | 19.96 (12.73–29.11) | 0.68 (0.00–1.02) | 0.67 (0.00–1.02) |
| *Ensemble* | | | | |
| *AUROC* | | | | |
| Standard | 93.70 (91.70–96.50) | 92.80 (89.80–95.60) | 100.00 (99.80–100.00) | 100.00 (100.00–100.00) |
| Enhanced | 94.17 (91.39–97.16) | 94.97 (93.18–96.81) | 99.41 (98.92–100.00) | 99.44 (99.12–100.00) |
| *FPR95 (%)* | | | | |
| Standard | 19.90 (11.20–29.10) | 32.70 (20.40–54.10) | 0.00 (0.00–0.00) | 0.00 (0.00–0.00) |
| Enhanced | 20.39 (11.71–29.59) | 26.12 (14.77–41.40) | 0.67 (0.00–1.02) | 0.67 (0.00–1.02) |

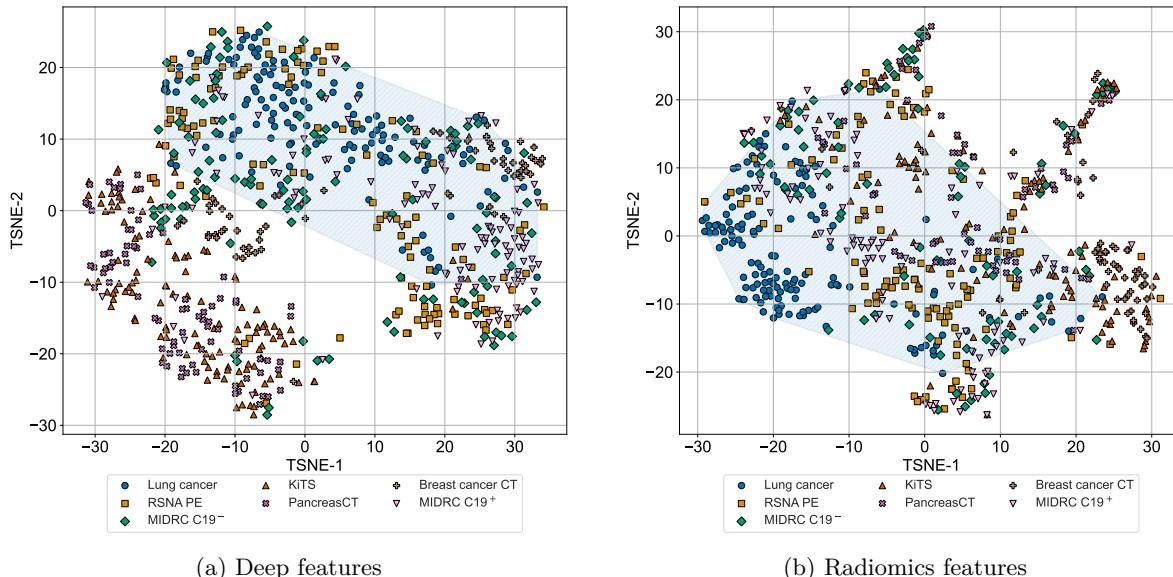

(a) Deep features         (b) Radiomics features

Figure 6: t-SNE projected embeddings showing dataset-wise separability of (a) deep features and (b) radiomics features. Results from one representative split (of 100), combining all datasets in a single visualization, are shown for brevity. The shaded blue regions indicate the convex hull of the ID dataset.

ID and anatomically distinct far-OOD datasets (KiTS, PancreasCT) and moderate overlap with near-OOD datasets, consistent with their increased detection difficulty. In contrast, radiomics features show poorly separated clusters with substantial overlap with ID scans, explaining their relatively lower accuracies (Table 1).

Across different backbones (Figure 15), all deep feature variants improved separability compared to the handcrafted radiomics features. SMIT-based backbones provided the most consistent separation, while

Table 4: MaxLogit scores across ID and OOD datasets within predicted tumor (positive class) regions, stratified into overall, boundary, and interior voxels.

| Dataset | Overall | Boundary | Interior |
|---|---|---|---|
| ID | $2.63 \pm 0.80$ | $1.13 \pm 0.35$ | $3.25 \pm 0.92$ |
| RSNA PE | $1.42 \pm 0.56$ | $0.74 \pm 0.25$ | $1.92 \pm 0.62$ |
| MIDRC C19$^-$ | $1.27 \pm 0.70$ | $0.82 \pm 0.41$ | $2.01 \pm 0.73$ |
| KiTS | $1.09 \pm 0.40$ | $0.85 \pm 0.34$ | $1.57 \pm 0.49$ |
| PancreasCT | $1.08 \pm 0.60$ | $0.66 \pm 0.32$ | $1.55 \pm 0.75$ |

Table 5: Classifier comparison on the pooled deep features. Point estimates of AUROC ($\uparrow$) and FPR95 ($\downarrow$) are reported over 100 matched-seed runs.

| Dataset | Linear Probing | MLP Classifier | Random forest |
|---|---|---|---|
| RSNA PE | 92.76 / 28.00 | 93.01 / 19.31 | **95.80 / 15.20** |
| MIDRC C19$^-$ | 92.36 / 29.46 | 92.88 / 27.88 | **93.30 / 25.50** |
| KiTS | 98.92 / 0.110 | 99.47 / 2.490 | **100.0 / 0.10** |
| PancreasCT | 99.94 / 0.460 | 99.25 / 0.800 | **100.0 / 0.00** |

differences between Swin UNETR variants highlighted the stronger influence of pretraining data relative to pretraining strategy.

### 5.3.2 RF-Deep extracts OOD data-agnostic and specific features for improved separability

We used SHAP (SHapley Additive exPlanations) (Lundberg & Lee, 2017) to identify which features drive RF-Deep and RF-Radiomics predictions, revealing how the two approaches differ in their decision-making mechanisms. Deep features (Figure 7) show broad, dataset-specific spread of important dimensions, with a small set of features reused across multiple datasets (Ft 65, Ft 104, Ft 131, Ft 179, Ft 319), suggesting they capture general distributional properties relevant to distinguishing ID from OOD scans. Signal magnitude scaled with OOD difficulty: far-OOD datasets showed wider, more strongly activated distributions consistent with their higher AUROC, while near-OOD datasets showed softer but structurally identical profiles. Additionally, the two unseen datasets produced SHAP profiles consistent with the seen near-OOD datasets, supporting the interpretation that RF-Deep learns pathology-relevant features rather than dataset-specific acquisition signatures.

In contrast, radiomics features (Figure 8) emphasized shape-based and texture-based descriptors, specifically the Gray-Level Co-occurrence Matrix (GLCM) and Gray-Level Run-Length Matrix (GLRLM) subsets, and while those lend interpretability compared to deep features, they were insufficient to capture the subtle differences across the datasets to accurately detect OOD.

### 5.3.3 Why logit-based spatial uncertainties are less accurate than RF-Deep?

To understand why logit-based OOD detectors were less accurate than RF-Deep, we analyzed MaxLogit spatial patterns within predicted tumor regions. We extracted boundary regions as a ring of 3 voxel thickness by performing erosion followed by dilation, each using a radius of 1 voxel. The interior contour following the erosion operation constituted the interior regions.

MaxLogit scores of ID and OOD scans show strong overall separation (Mann-Whitney U tests, all p < 0.001, effect sizes ranging from r = $-0.75$ for MIDRC C19$^-$ to r = $-0.90$ for KiTS) due to higher overall MaxLogit scores for ID scans than individual OOD datasets (Table 4). However, the scores have substantially overlapping ranges with ID (ID: 0.12–4.33, RSNA PE: 0.09–3.67, MIDRC C19$^-$: 0.30–2.70).

Concretely, interior voxels showed consistently higher MaxLogit scores than the boundary voxels in both ID and OOD predictions. For ID scans, the interior scores were roughly three times higher than boundary scores, and this ratio persists across all OOD datasets despite their lower absolute scores. Paired, two-sided Wilcoxon signed-rank tests comparing interior and boundary scores within each dataset confirmed the universality of this spatial pattern (p < 0.001 for all datasets, effect sizes r ≈ -1.0), indicating that the boundary–interior relationship is preserved regardless of ID or OOD status, further limiting their discriminative power.

Figure 9 provides a visualization of the MaxLogit spatial uncertainties for four representative scans. The density-normalized histograms depict the distribution of MaxLogit values in the interior and boundary regions of the detected lesions. The interior and boundary regions show very good separation for the ID case (row 1) and sufficiently different spatial distribution for the second case (row 2), enabling correct detection of OOD for this case. However, the third case (row 3) presents a challenge wherein the spatial uncertainty

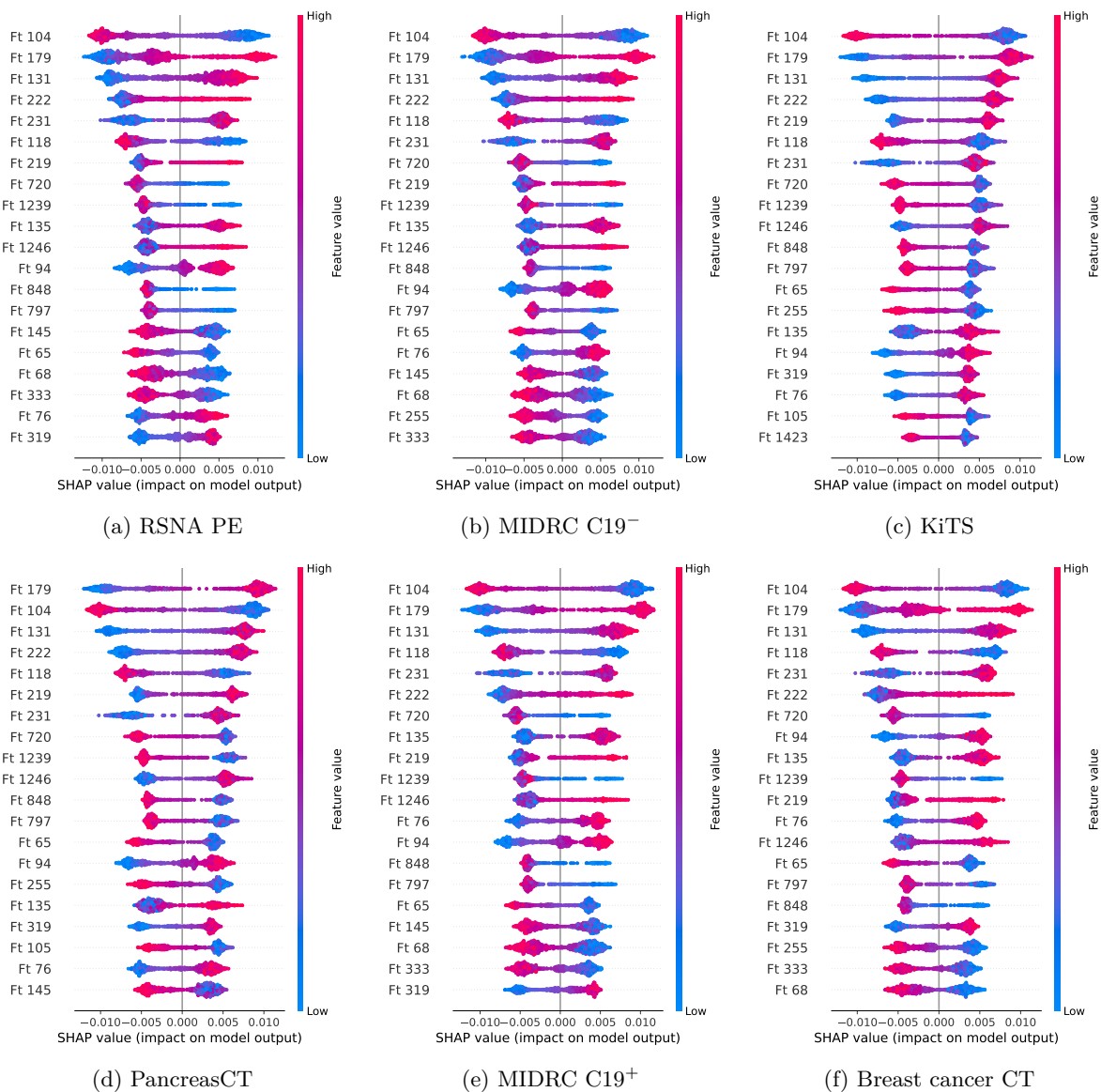

Figure 7: SHAP feature importance for ensemble RF-Deep across all six OOD datasets. Ft 104 and Ft 179 rank as the top two features across all datasets regardless of OOD category or acquisition provenance. Of note, models were trained exclusively on the four standard OOD datasets (RSNA, MIDRC C19$^-$, KiTS, PancreasCT); the two external datasets (MIDRC C19$^+$, Breast cancer CT) were held out entirely and used only for evaluation, demonstrating that the learned feature importance generalizes to unseen OOD distributions.

distribution closely resembles that of the ID case, resulting in incorrect classification. In contrast, RF-Deep correctly identified both OOD scans, demonstrating its superior ability to capture contextual features beyond local spatial patterns. Of note, both RF-Deep and MaxLogit-based OOD prediction was wrong for the fourth case (row 4), presenting a limiting case for either method.

An additional limitation of logit-based methods is their inherent scale invariance: MaxLogit scores reflect local appearance characteristics independent of lesion size as shown by the distributions for lesions of varying sizes in Figure 9. This is because MaxLogit computation treats voxels identically based on their pre-softmax logit values, ignoring volumetric context that could differentiate true tumors from anatomically

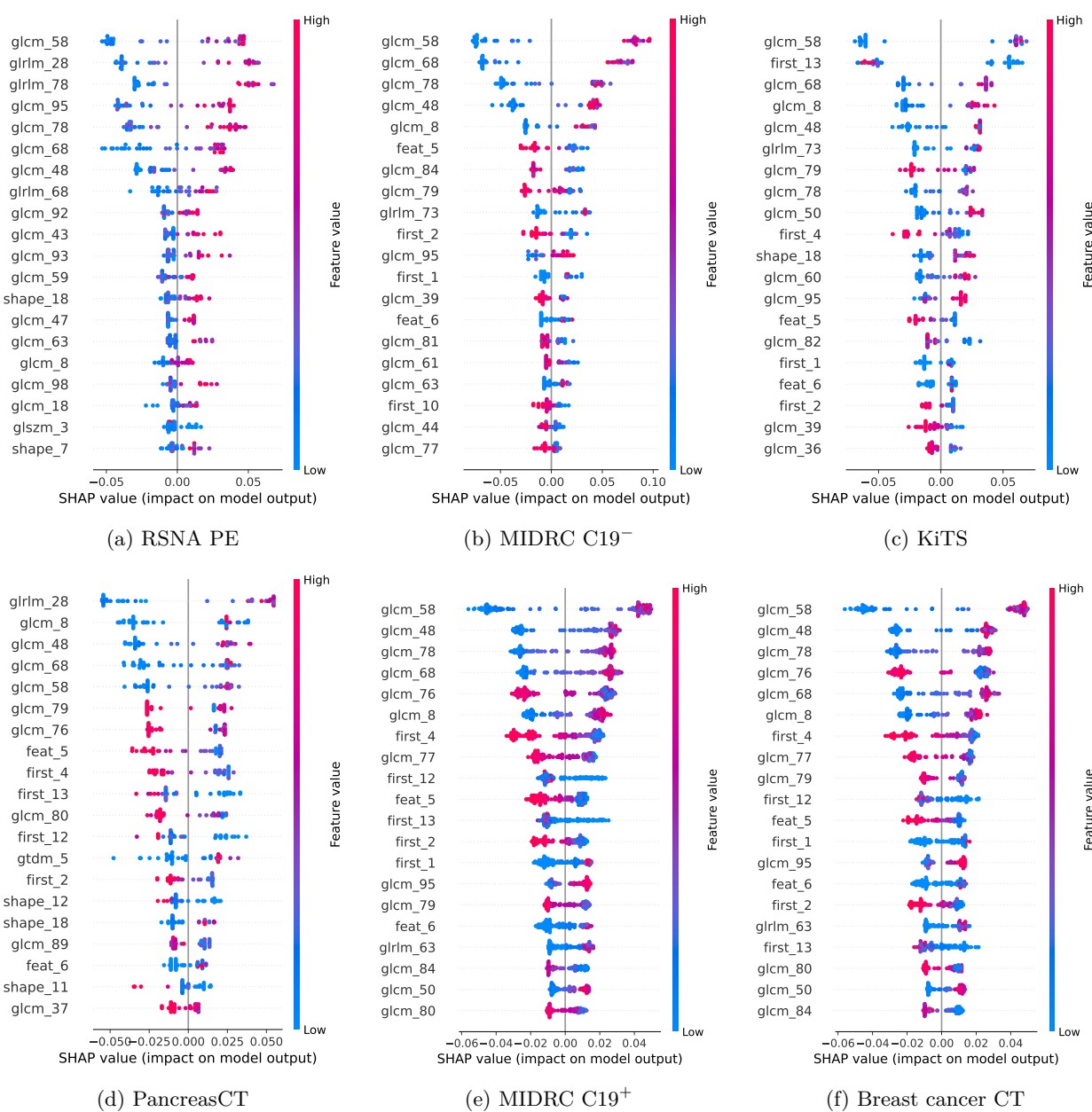

Figure 8: SHAP feature importance for RF-Radiomics across OOD datasets using segmentations from the SMIT-pretrained model. RF-Radiomics primarily relies on GLCM and GLRLM texture descriptors.

implausible detections. This scale invariance poses a critical bottleneck for OOD detection, as many false-positive detections on OOD scans manifest as small, scattered regions with locally high confidence but globally inconsistent spatial distributions. RF-Deep, on the other hand, aggregates features from regions encompassing detections, reducing the impact of tumor size variations. Restricting logit-based aggregation to the same tumor-anchored ROIs used by RF-Deep did not improve their performance (Table 11), confirming that the advantage of RF-Deep stems from its feature representation rather than its spatial aggregation strategy.

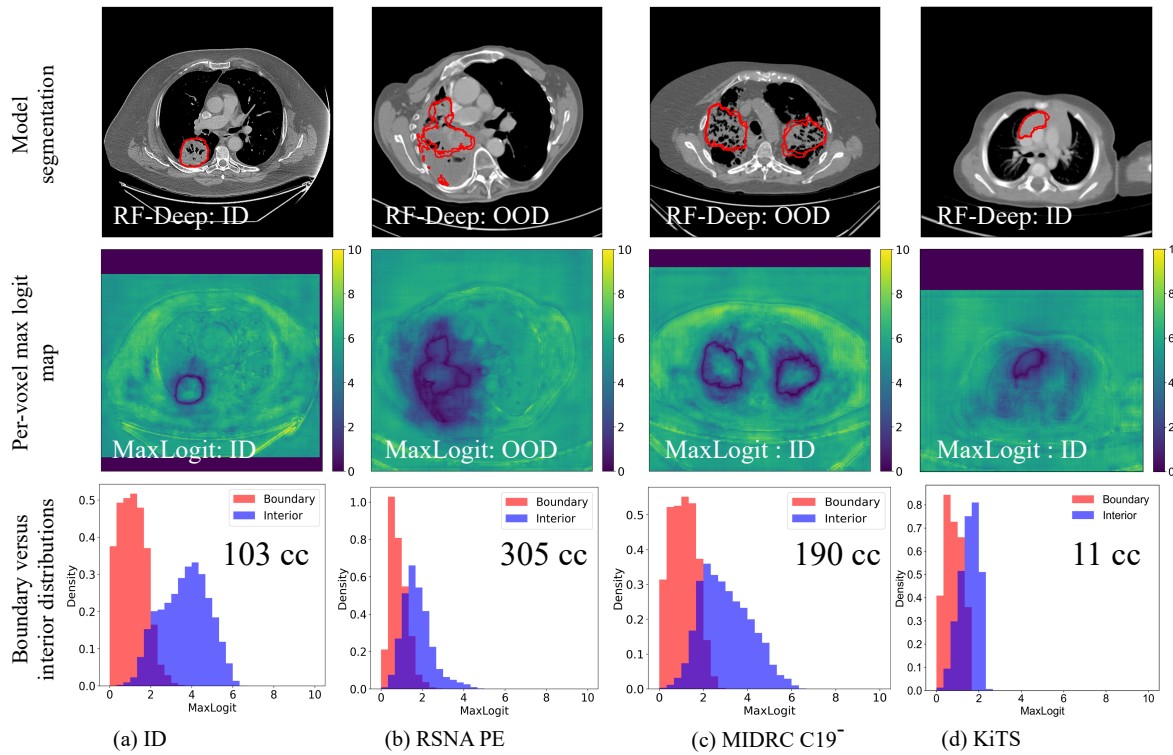

Figure 9: Representative MaxLogit visualizations for (a) segmentation predictions, (b) spatial heatmaps, and (c) boundary vs. interior distributions with predicted tumor volumes. Violet regions indicate unevaluated voxels due to foreground thresholding.

Table 6: OOD detection performance across deployment strategies. Point estimates and 95% bootstrap confidence intervals over 100 matched-seed runs are reported.

| Strategy | AUROC ↑ | | | | FPR95 (%) ↓ | | | |
|---|---|---|---|---|---|---|---|---|
| | RSNA PE | MIDRC C19⁻ | KiTS | PancreasCT | RSNA PE | MIDRC C19⁻ | KiTS | PancreasCT |
| Dataset-specific | 95.80 | 93.30 | 100.00 | 100.00 | 15.20 | 25.50 | 0.10 | 0.00 |
| | (92.60–98.50) | (89.90–96.00) | (99.90–100.00) | (100.00–100.00) | (7.10–23.50) | (16.80–36.20) | (0.00–1.00) | (0.00–0.00) |
| Ensemble | 93.70 | 92.80 | 100.00 | 100.00 | 19.90 | 32.70 | 0.00 | 0.00 |
| | (91.70–96.50) | (89.80–95.60) | (99.80–100.00) | (100.00–100.00) | (11.20–29.10) | (20.40–54.10) | (0.00–0.00) | (0.00–0.00) |
| LODO | 91.20 | 91.00 | 99.80 | 100.00 | 25.70 | 43.20 | 0.70 | 0.20 |
| | (85.50–95.30) | (84.40–94.80) | (99.30–100.00) | (99.70–100.00) | (17.30–36.70) | (23.50–54.10) | (0.00–4.10) | (0.00–2.00) |

## 5.4 Framework generalization and robustness

### 5.4.1 Effect of classifier selection

Linear probing and multilayer perceptron (MLP) classifiers, both implemented with default settings from scikit-learn, provided reasonable separability on the far-OOD datasets (Table 5), but their performance degrades notably on the near-OOD datasets (AUROC ≈ 92–93%, FPR95 ≈ 28–30%). In contrast, RFs achieve the best performance across all datasets, highlighting the advantage of tree-based non-linear classifiers for capturing complex decision boundaries, particularly when distribution shifts are subtle (Figure 6a). The perfect performance on KiTS and PancreasCT (AUROC = 100%, FPR95 ≈ 0%) reflects the very easy nature of far-OOD detection when anatomical regions differ substantially from the training distribution.

Table 7: Acquisition factor robustness: leave-one-factor-out within the ID cohort. False OOD rate (%) and OOD detection AUROC with 95% CIs over 100 matched-seed runs.

| Factor | Held-out | Dataset-specific | | Ensemble | |
|---|---|---|---|---|---|
| | | False OOD % | AUROC | False OOD % | AUROC |
| *Scanner manufacturer* | | | | | |
| | GE | 9.07 (6.19–12.62) | 97.95 (96.83–98.82) | 1.31 (0.00–4.31) | 95.67 (93.95–96.97) |
| | Siemens | 10.15 (4.35–18.48) | 97.98 (96.58–98.94) | 2.52 (0.00–8.70) | 96.19 (94.03–97.63) |
| *Contrast status* | | | | | |
| | Contrast | 8.75 (5.57–14.72) | 98.14 (95.47–99.03) | 0.87 (0.00–3.53) | 96.31 (92.31–97.96) |
| | Non-contrast | 14.70 (9.72–20.61) | 96.69 (95.17–97.81) | 8.39 (0.00–18.52) | 93.71 (91.45–95.57) |
| *Reconstruction kernel* | | | | | |
| | Recon 1 | 43.74 (34.56–54.83) | 91.60 (87.55–94.36) | 42.68 (14.71–70.59) | 85.87 (79.67–90.66) |
| | Recon 2 | 3.74 (1.32–11.18) | 99.13 (98.50–99.58) | 1.16 (0.00–2.63) | 98.32 (97.18–99.23) |
| | Recon 3 | 3.66 (0.78–11.35) | 98.47 (96.29–99.73) | 1.45 (0.00–6.25) | 96.33 (91.89–98.95) |

### 5.4.2 Effect of deployment strategy

We additionally evaluated RF-Deep under the leave-one-dataset-out (LODO) strategy, which trains on ID plus three OOD datasets and evaluates on the held-out fourth, directly measuring generalization to in-house unseen OOD datasets (Table 6).

Performance patterns on near-OOD datasets (RSNA PE and MIDRC C19$^-$) revealed important insights on within-anatomy generalization. The ensemble strategy approached dataset-specific performance while maintaining the ability to generalize to new test domains. In contrast, the LODO strategy showed reduced performance, particularly on near-OOD datasets, indicating that exposure to diverse OOD distributions improves within-anatomy generalization. All strategies achieve near-perfect performance on far-OOD datasets, reflecting their lower difficulty.

### 5.4.3 Robustness of RF-Deep to acquisition and dataset biases

To assess whether RF-Deep relies on acquisition-specific cues, we performed leave-one-factor-out experiments across scanner manufacturer, contrast usage, and reconstruction kernel (Table 7). False OOD rates remained low for scanner and contrast variations, with consistently high AUROC, indicating robustness to acquisition differences. Elevated false OOD rates were observed for Recon 1 kernels, that were attributable to correlated acquisition factors rather than kernel-specific bias. The threshold sensitivity analysis (Figure 14) over $\tau \in \{0.3, 0.4, 0.5, 0.6, 0.7\}$ confirmed that scanner and contrast holdouts remained stable, whereas Recon 1 remained the only consistently elevated subgroup.

We additionally examined reconstruction kernel metadata in the RSNA PE dataset and identified at least 10 distinct kernel values across 1,225 studies (e.g., STANDARD, B30f, FC08-H, SOFT), confirming substantial acquisition heterogeneity within this near-OOD dataset. RF-Deep's consistent detection performance across this diversity further suggests that detection is not attributable to kernel homogeneity within the OOD datasets.

### 5.4.4 Inference timing

RF-Deep adds minimal computational overhead to the segmentation pipeline (Table 10). On a single NVIDIA A40 GPU, segmentation inference (sliding window) dominated runtime at 18.4 secs per scan for the SMIT backbone. RF-Deep's feature extraction and classification added only 0.31 secs (dataset-specific) to 0.51 secs (ensemble), corresponding to overheads of 1.7% and 2.8%, respectively. The RF classifier runs completely on CPU and introduces no additional trainable parameters to the segmentation backbone. By comparison, the

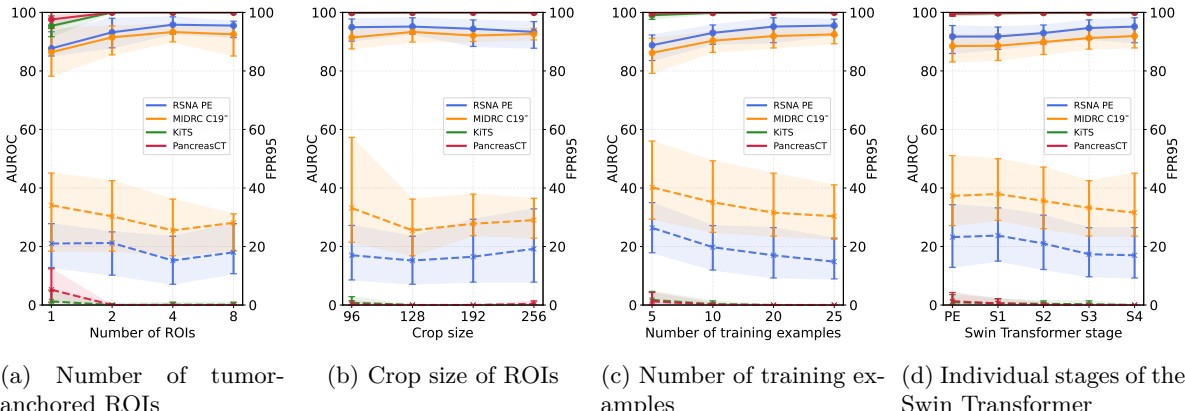

(a) Number of tumor-anchored ROIs

(b) Crop size of ROIs

(c) Number of training examples

(d) Individual stages of the Swin Transformer

Figure 10: Ablation studies showing the sensitivity of RF-Deep for OOD detection across four datasets. Point estimates and 95% bootstrap confidence intervals over 100 runs with matched seeds are displayed.

Swin UNETR backbone took 29.8 secs per scan due to its smaller ROI size (requiring more sliding windows) while the RF-Deep overhead remained comparable at 0.20 secs (dataset-specific) to 0.51 secs (ensemble). Finally, RF-Radiomics required 20–120 secs of additional radiomics extraction per scan.

## 5.5 Design choices and sensitivity analysis

To understand the design choices of our proposed OOD detector, we conducted a series of controlled ablations on the number of ROIs, their crop sizes, the extent of labeled (supervised) data, and the encoder stages used for feature extraction. Figure 10a shows that OOD detection benefits from richer local context around predicted tumor regions, but only up to a degree. With $n = 1$, performance is unstable, reflected in wider confidence intervals and elevated FPR95, suggesting that a single tumor-anchored crop may miss informative variation in tumor surroundings. Increasing the number of ROIs from one to four improved performance on near-OOD scans, indicating that additional samples may better disambiguate similarly appearing tumor (ID) from non-cancerous structures. Beyond four ROIs, gains plateau or occasionally regress (e.g., MIDRC C19$^-$), likely due to noise injection from ROIs sampling redundant or low-value crops. Far-OOD datasets did not exhibit performance gains with increasing number of ROIs. Therefore, we used $n = 4$ to capture meaningful contextual diversity while maintaining computational efficiency.

Figure 10b analyzes the effect of tumor-anchored 3D crop size on OOD detection performance. Performance remained relatively stable on the far-OOD datasets across crop sizes, with slight preference for $128 \times 128 \times 128$. On the near-OOD datasets, both smaller ($< 128$) and larger ($> 128$) ROIs led to performance degradation, suggesting that the optimal contextual window balances tumor-centered information with surrounding anatomical context.

Figure 10c shows that RF-Deep's OOD detection performance improved with the number of labeled examples, using equal numbers of ID and OOD scans (5/5, 10/10, up to 20/20, corresponding to less than 30% of available data), after which performance plateaued. With only 20 ID and 20 OOD labeled scans, RF-Deep achieved AUROC $> 90$ and FPR95 $< 30\%$ on average across all OOD datasets. Performance gains were more substantial on the near-OOD datasets than the far-OOD datasets, indicating that detection on subtle shifts in the same anatomical region is harder, requiring more examples than large distribution shifts across different anatomical regions. Variance of the AUROC decreases consistently as more samples are added, with the strongest variance reduction observed on near-OOD datasets.

Figure 10d shows individual Swin Transformer stage contributions. Far-OOD datasets were relatively insensitive to individual stage selection (achieving AUROC $> 98$ even when using features from a single stage), whereas the near-OOD counterparts showed strong sensitivity to early stages, with Stage 1 and Stage 2 features being particularly important (AUROC $\approx 88$–$90$ individually). Concatenating features across all stages yielded the best performance, confirming that hierarchical representations are essential for robust

OOD detection. This finding aligns with the SHAP analysis (Section 5.3.2), which showed that RF-Deep adaptively leverages features from different abstraction levels depending on the OOD scenario.

Figure 11 compares crop strategies using identical encoder features and RF classifiers. Even without tumor guidance, the hierarchical encoder features provided strong OOD discrimination (AUROC $\geq 92$ on all near-OOD datasets), confirming that the SSL-pretrained-then-finetuned backbone is the primary driver of detection. Tumor anchoring provided consistent additional gains, most evident on the hardest near-OOD dataset (RSNA PE: AUROC improved from 93.70 to 95.80; FPR95 reduced from 20.0% to 15.2%) by focusing feature extraction on the regions most relevant to the segmentation task's decision boundary. While the AUROC improvements are moderate on the evaluated datasets, tumor anchoring reduces FPR95 at high-sensitivity operating points and ensures that OOD detection remains grounded in task-relevant regions, which is essential when scans vary substantially in field of view or when the clinical rationale for flagging must relate to the segmentation output.

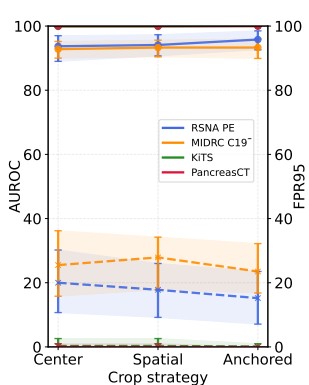

Figure 11: Effect of crop strategy. Center: single body-center crop; Spatial: 4 random crops; Anchored: 4 tumor-anchored crops (proposed).

## 6    Discussion and conclusion

Effective near-OOD detection requires both discriminative features and learned decision boundaries, which form the foundation of RF-Deep's design. Despite using identical deep features, the unsupervised Mahalanobis distance (no OOD scans) performed substantially worse than RF-Deep on the near-OOD datasets, demonstrating that outlier exposure and non-linear feature selection are essential for distinguishing diseases within the same anatomical site. RF-Deep improved over logit-based methods, indicating that over-reliance on the ID model to extract confidence boundaries may be unreliable in OOD scans. Although this issue could be addressed to an extent through confidence calibration, prior studies show limitations of such methods (Larrazabal et al., 2023; Mehrtash et al., 2020). RF-Deep overcomes this limitation by combining SSL-pretrained backbones for segmentation with lightweight random forest classifiers, providing a lightweight, post-hoc mechanism to extend existing task-specific DL models for OOD detection. Our ablation analysis further revealed that the SSL-pretrained encoder features are the dominant contributor to OOD detection, with tumor anchoring providing incremental but operationally meaningful gains concentrated at high-sensitivity operating points (Section 5.5, Figure 11). These results suggest that RF-Deep's framework can potentially benefit other sufficiently expressive pretrained backbones, with task-specific anchoring offering additional refinement for the most challenging near-OOD scenarios, although this remains to be verified beyond the Swin Transformer family evaluated in this paper.

**Potential biases and mitigation.** To mitigate data-specific biases, we employed: (a) SSL pretraining on 10,432 diverse CT scans encompassing multiple institutions and scanner manufacturers, (b) separate training and testing datasets from different patient datasets providing external validation, and (c) disaggregated evaluation across scanner manufacturer, reconstruction kernels, and contrast agent presence (Figure 3b).

**Limitations.** Our framework is designed for scan-level OOD detection. As a result, false detection instances in the ID scan are not explicitly addressed and remain an area for future work. In practice, this limitation could be mitigated by integrating RF-Deep with complementary scan-level quality control (using [CLS] tokens) or uncertainty estimation mechanisms, which we plan to explore in future work. Our current approach was applied to thoracic cancers detected on CTs. Extension to other disease sites and more challenging imaging modalities such as magnetic resonance imaging which exhibit greater variability, is planned future work. While the framework is not inherently modality-specific, its robustness in these settings requires further empirical validation. Leave-one-factor-out experiments confirmed robustness to scanner manufacturer and contrast variation, though the Recon 1 kernel subgroup exhibited elevated false OOD rates under maximally adversarial holdout conditions, likely due to correlated acquisition factors rather than kernel-specific bias. Additionally, RF-Deep requires a modest set of representative outlier examples for training, which may not be available in early deployment settings where failure modes have not yet been observed. However, the

framework is compatible with incremental updates as new failure cases are encountered, allowing the OOD detector to improve over time with minimal additional supervision.

In summary, we introduced RF-Deep, a lightweight post-hoc OOD detection framework for lung tumor segmentation models that leverages hierarchical encoder features and random forest classifiers with limited outlier exposure. RF-Deep can be integrated into segmentation pipelines under multiple deployment regimes. When likely failure modes are known a priori, dataset-specific training with representative outlier exposure can yield stronger discrimination. In settings where deployment shifts are uncertain, an ensemble trained across diverse exposure cohorts provides more stable rejection performance. Across near-OOD and far-OOD datasets, including two blinded validation datasets, RF-Deep consistently outperformed logit-based, radiomics-based, and unsupervised feature-space alternatives. Future work will explore extension to multi-label segmentations and integration of the absent-prediction signal for improved robustness.

**Broader Impact Statement**

This work addresses a critical gap between research-grade AI performance and clinical deployment requirements for medical image segmentation. By providing reliable OOD detection, RF-Deep can help prevent confidently incorrect predictions from reaching clinical workflows, potentially reducing the risk of inappropriate downstream decisions from automated systems that fail silently on out-of-distribution inputs. However, no OOD detection method is perfect; the evaluated datasets capture key failure modes, but covering all possible diseases is impractical. Importantly, RF-Deep operates conditional on a non-empty segmentation prediction; cases with no predicted foreground may require independent scan-level quality control or anomaly detection mechanisms. These considerations highlight that RF-Deep is best viewed as a modular risk-aware screening component rather than a standalone safety guarantee.

**Acknowledgments**

This research was partially supported by the NCI R01CA258821 and the Memorial Sloan Kettering Cancer Center Support Grant/Core Grant NCI P30CA008748. We sincerely thank Jue Jiang, Chloe Min Seo Choi, and Nishant Nadkarni for their thoughtful feedback, critical review of the manuscript, and helpful discussions, which significantly strengthened this work.

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

## A  Pretraining strategies descriptions

SMIT (Jiang et al., 2022) consists of masked voxel token prediction, masked patch token distillation, and image token distillation. Masked voxel token prediction is a dense regression task of the intensities within masked patches and is optimized by minimizing the $\ell_1$ norm between the predicted and ground-truth patches, akin to SimMIM (Xie et al., 2022). Self-distillation (Tarvainen & Valpola, 2017) was required for the next two tasks and was performed using an exponentially moving average (EMA) teacher model with identical architecture as the student, similar to iBOT (Zhou et al., 2022). Masked patch token distillation aligns the voxel token representations learned by the student with those extracted by the teacher from unmasked sequences, optimized with a temperature-scaled cross-entropy loss. For image token distillation, the cross-entropy between global volume embeddings of the student and teacher was minimized. After pretraining, the teacher was discarded and the student encoder was retained for fine-tuning.

Similarly, Swin UNETR (Tang et al., 2022) employs three pretraining tasks. The primary pretext task is masked volume reconstruction, where a proportion of 3D input patches are randomly masked and the model is trained to regress their voxel intensities from visible context. Unlike SimMIM, which leverages positional encodings to aid masked token prediction, this reconstruction task provided no explicit positional information, forcing the model to learn semantically plausible inpainting from local context. 3D rotation prediction enforces invariance by classifying the degree of rotation applied along the $z$-axis. Finally, contrastive coding encouraged representation learning by maximizing cosine similarity between augmented views of the same volume while minimizing similarity for different volumes. Unlike SMIT, which requires an EMA teacher, Swin UNETR used only a student model. The pretrained encoder was directly retained for fine-tuning.

Table 8: Feature-space OOD detection approaches used with MD-Deep. AUROC and FPR95 (%) with 95% CIs over 100 matched-seed runs.

| Method | RSNA PE | MIDRC C19$^-$ | KiTS | PancreasCT |
|---|---|---|---|---|
| *AUROC* | | | | |
| MD-Deep | 87.30 (83.90–91.10) | 84.50 (79.40–89.00) | 99.20 (97.90–99.80) | 99.20 (98.20–100.00) |
| MD + ReAct | 83.25 (78.95–87.41) | 80.45 (74.55–86.63) | 98.68 (96.55–99.62) | 98.27 (95.76–99.67) |
| MD + ASH | 73.34 (69.23–78.65) | 76.68 (71.70–81.67) | 97.93 (95.80–99.35) | 97.24 (93.90–99.06) |
| X-Mahalanobis | 85.44 (82.09–88.94) | 81.26 (75.45–87.32) | 98.55 (96.85–99.71) | 98.59 (97.06–99.93) |
| RF-Deep | 95.80 (92.60–98.50) | 93.30 (89.90–96.00) | 100.00 (99.90–100.00) | 100.00 (100.00–100.00) |
| *FPR95* | | | | |
| MD-Deep | 45.10 (30.60–60.30) | 58.50 (38.20–77.60) | 3.10 (0.00–9.20) | 2.80 (0.00–7.70) |
| MD + ReAct | 54.62 (40.77–67.88) | 64.76 (43.34–85.23) | 5.79 (1.02–11.76) | 6.39 (1.51–13.27) |
| MD + ASH | 83.01 (69.34–92.86) | 75.73 (61.71–90.87) | 8.45 (2.04–20.41) | 10.61 (3.55–26.05) |
| X-Mahalanobis | 51.16 (40.82–63.27) | 63.29 (44.85–83.67) | 3.88 (0.00–9.18) | 4.32 (1.02–12.24) |
| RF-Deep | 15.20 (7.10–23.50) | 25.50 (16.80–36.20) | 0.10 (0.00–1.00) | 0.00 (0.00–0.00) |

## B  Logit-based approaches formulae

The tumor segmentation problem can be broken down into a per-voxel binary classification problem, in which we mean-aggregate the scores of the positive class to summarize the scan. In-distribution (ID) scans tended to yield more confident outputs, whereas out-of-distribution (OOD) scans produced less confident or spurious predictions. Following the convention of treating OOD as the positive class (Hendrycks & Gimpel, 2017; Choi et al., 2024), we negated each score so that higher values correspond to greater likelihood of being OOD. Let $f^i(x)$ denote the logit for class $i \in \{0,1\}$ at voxel $x$, and define the softmax score as

$$S^i_{\text{softmax}}(x) \;=\; \frac{\exp\big(f^i(x)\big)}{\sum_{j=0}^1 \exp\big(f^j(x)\big)}.$$

We denote the set of voxels predicted as positive (tumor) by

$$V_+ \;=\; \big\{(h,w,d)\,\big|\,\hat{y}_{h,w,d}=1\big\},$$

where $\hat{y}_{h,w,d}$ is the class prediction at voxel $(h,w,d)$. Each logit-based score aggregates over $V_+$:

- **MaxSoftmax** (Hendrycks & Gimpel, 2017): $S = -\frac{1}{|V_+|} \sum_{(h,w,d)\in V_+} S^1_{\text{softmax}}(x_{h,w,d})$.

- **MaxLogit** (Hendrycks et al., 2022): $S = -\frac{1}{|V_+|} \sum_{(h,w,d)\in V_+} f^1(x_{h,w,d})$.

- **Energy** (Liu et al., 2020): $S = -\frac{1}{|V_+|} \sum_{(h,w,d)\in V_+} \log\Big( \sum_{i=0}^1 \exp\big(f^i(x_{h,w,d})\big)\Big)$.

## C  Additional feature-space OOD detection approaches with MD-Deep

We evaluated three recent post-hoc feature-space methods adapted to our 3D segmentation setting: ReAct Sun et al. (2021), ASH Djurisic et al. (2022), and X-Mahalanobis Wei et al. (2025). All methods were applied to the same pre-extracted multi-scale encoder features used by MD-Deep and paired with Mahalanobis distance scoring.

From Table 8, both ReAct and ASH reduced AUROC relative to standard MD-Deep on the near-OOD datasets (ASH: $-13.96$ on RSNA PE, $-7.82$ on MIDRC C19$^-$; ReAct: $-4.05$ on both). X-Mahalanobis

Table 9: Radiomics features definitions as per IBSI guidelines.

| ID | Feature | ID | Feature |
|---|---|---|---|
| *General features* | | | |
| *feat*_5 | maxIntensityInterpReseg | *feat*_6 | minIntensityInterpReseg |
| | | | |
| *Shape features* | | | |
| *shape*_7 | max2dDiameterSagittalPlane | *shape*_11 | volume |
| *shape*_12 | filledVolume | *shape*_18 | surfToVolRatio |
| | | | |
| *First-order features* | | | |
| *first*_1 | min | *first*_2 | max |
| *first*_4 | range | *first*_10 | entropy |
| *first*_12 | energy | *first*_13 | totalEnergy |
| | | | |
| *Gray-Level Co-occurrence Matrix (GLCM) features* | | | |
| *glcm*_8 | jointEntropy_3D_StdDev | *glcm*_18 | jointAvg_3D_StdDev |
| *glcm*_36 | sumEntropy_3D_avg | *glcm*_37 | sumEntropy_3D_Median |
| *glcm*_39 | sumEntropy_3D_Min | *glcm*_43 | contrast_3D_StdDev |
| *glcm*_44 | contrast_3D_Min | *glcm*_47 | invDiffMom_3D_Median |
| *glcm*_48 | invDiffMom_3D_StdDev | *glcm*_58 | invDiff_3D_StdDev |
| *glcm*_59 | invDiff_3D_Min | *glcm*_60 | invDiff_3D_Max |
| *glcm*_61 | invDiffNorm_3D_avg | *glcm*_63 | invDiffNorm_3D_StdDev |
| *glcm*_68 | invVar_3D_StdDev | *glcm*_76 | diffEntropy_3D_avg |
| *glcm*_77 | diffEntropy_3D_Median | *glcm*_78 | diffEntropy_3D_StdDev |
| *glcm*_79 | diffEntropy_3D_Min | *glcm*_80 | diffEntropy_3D_Max |
| *glcm*_81 | diffVar_3D_avg | *glcm*_82 | diffVar_3D_Median |
| *glcm*_84 | diffVar_3D_Min | *glcm*_89 | diffAvg_3D_Min |
| *glcm*_92 | corr_3D_Median | *glcm*_93 | corr_3D_StdDev |
| *glcm*_95 | corr_3D_Max | *glcm*_98 | clustTendency_3D_StdDev |
| | | | |
| *Gray-Level Run-Length Matrix (GLRLM) features* | | | |
| *glrlm*_28 | runLengthNonUniformityNorm_3D_StdDev | *glrlm*_68 | grayLevelVariance_3D_StdDev |
| *glrlm*_73 | runLengthVariance_3D_StdDev | *glrlm*_78 | runEntropy_3D_StdDev |
| | | | |
| *Gray-Level Size Zone Matrix (GLSZM) features* | | | |
| *glszm*_3 | grayLevelNonUniformity_3D | | |
| | | | |
| *Gray-Tone Difference Matrix (GTDM) features* | | | |
| *gtdm*_5 | strength_3D | | |

provided no improvement over standard MD-Deep on any dataset, consistent with our baseline already operating on concatenated multi-scale features. The 8–10 AUROC point gap between the best unsupervised method and RF-Deep on near-OOD datasets persisted regardless of feature-space transformation, reinforcing that outlier exposure and non-linear classification are the critical differentiators.

## D   Additional results

This section provides additional quantitative and qualitative results that support the findings presented in the paper.

- Table 9 lists the mapping from feature IDs used for reporting results, mentioned briefly in Sections 4.4, 5.2, and 5.3.

- Table 10 quantifies the inference time required to process scans across the SMIT and Swin UNETR segmentation models, discussed briefly in Section 5.4.4.

- Figure 12 presents the disaggregated analysis of segmentation performance across imaging variations, discussed briefly in Section 5.1 (Figure 3).

Table 10: Per-scan runtime for OOD detection, averaged over 50 LRAD scans on a single GPU with CPU-side classifier inference. All timing measurements were performed on an NVIDIA A40 GPU and dual Intel Xeon Platinum 8358 CPUs.

| Method / Stage | SMIT (s) | Swin UNETR (s) |
|---|---|---|
| Segmentation inference (shared) | 18.39 | 29.79 |
| Logit post-processing (MaxSoftmax / MaxLogit / Energy) | 0.50 | 0.50 |
| RF-Deep feature extraction (4 ROIs) | 0.23 | 0.12 |
| RF-Deep classifier, dataset-specific (1 RF, CPU) | 0.08 | 0.08 |
| RF-Deep classifier, ensemble (4 RFs, CPU) | 0.31 | 0.40 |
| RF-Radiomics overhead ($\sim$) | 20–120 | 20–120 |
| MD-Deep overhead, dataset-specific ($\sim$) | 0.28 | 0.20 |
| RF-Deep overhead, dataset-specific | 0.31 | 0.20 |
| RF-Deep overhead, ensemble | 0.51 | 0.51 |

- Figure 13 supports the discussion in Section 5.2 on MD-Deep, demonstrating that the standard formulation outperforms all variants.

- Figure 14 presents the threshold sensitivity analysis across acquisition factors, supporting the robustness discussion in Section 5.4.

- Table 12 summarizes OOD performance for all methods (five additional pretraining strategies on six methods), reinforcing the findings in Section 5.2.

- Table 11 compares logit-based methods under ROI-restricted aggregation, discussed in Section 5.2.

- Figure 15 shows that while all deep feature variants improve separability relative to radiomics, SMIT-based backbones yield the most consistent clustering across both near- and far-OOD datasets, complementing Figure 6a and discussions in Sections 5.2 and 5.3.1.

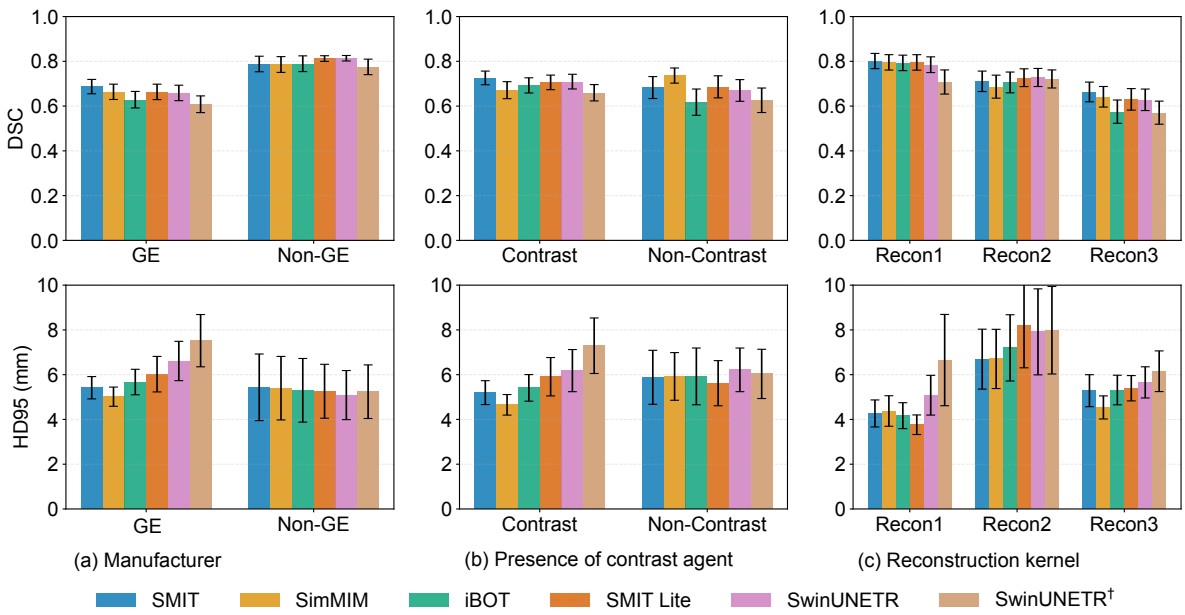

Figure 12: Segmentation performance across backbones under imaging variations (scanner type, contrast, reconstruction kernel) on the ID dataset. DSC and HD95 are reported.

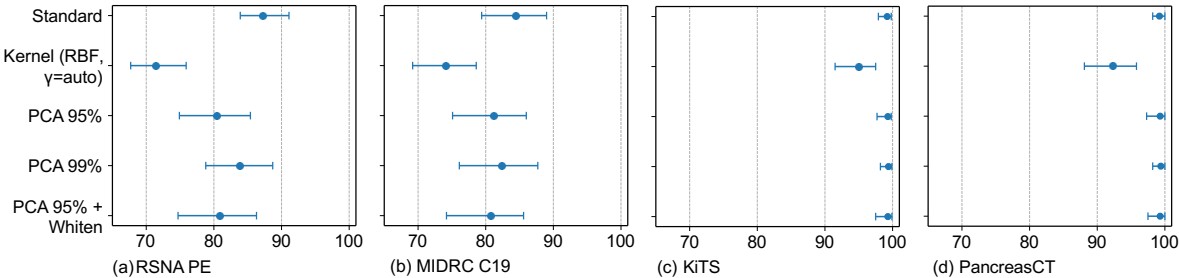

Figure 13: Ablation of Mahalanobis distance-based variants for OOD detection using AUROC over 100 matched-seed runs.

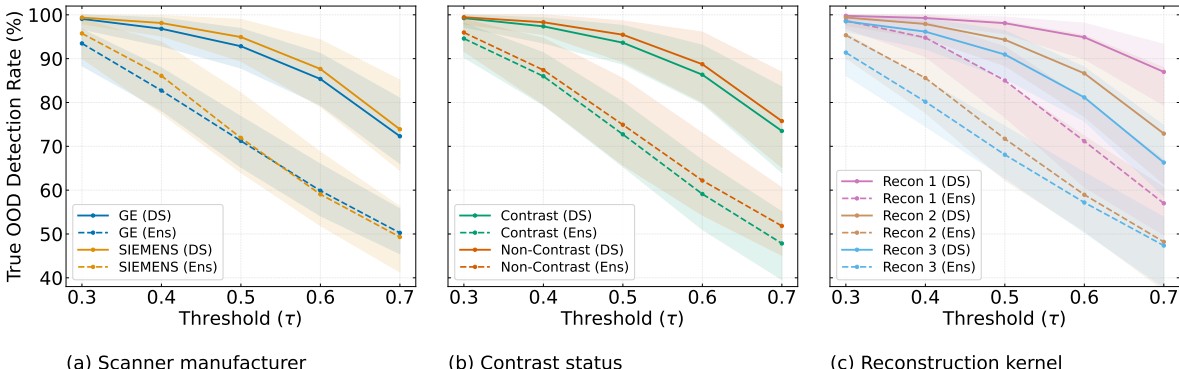

Figure 14: Threshold sensitivity analysis for acquisition factor robustness across $\tau \in 0.3, 0.4, 0.5, 0.6, 0.7$. Shaded regions show 95% CIs over 100 matched-seed runs. False OOD rates (not shown) decrease monotonically with $\tau$; at the operating threshold $\tau = 0.5$, false OOD rates remained below 15% for manufacturer and contrast factors, though reconstruction kernel holdout exhibited higher rates (up to 43%) driven by the Recon 1 subgroup.

Table 11: Comparison of logit-based methods with original (all-voxel) and ROI-restricted aggregation on the primary benchmark cohorts. Metrics are reported as percentages with 95% bootstrap confidence intervals.

| Method | Aggregation | RSNA PE | MIDRC C19$^-$ | KiTS | PancreasCT |
|---|---|---|---|---|---|
| *AUROC* | | | | | |
| MaxSoftmax | All voxels | 88.61 (84.62–92.36) | 86.57 (81.79–90.14) | 95.67 (93.41–98.06) | 94.61 (92.61–97.49) |
| | ROI-restricted | 86.43 (82.05–90.33) | 87.22 (81.91–91.42) | 95.57 (93.07–97.63) | 89.28 (85.66–93.43) |
| MaxLogit | All voxels | 88.77 (85.09–92.43) | 89.31 (85.05–93.06) | 95.89 (93.58–98.44) | 93.53 (90.17–96.59) |
| | ROI-restricted | 87.38 (83.06–91.14) | 89.18 (84.72–92.75) | 95.79 (92.92–97.73) | 91.63 (88.32–95.23) |
| Energy | All voxels | 88.59 (84.85–92.24) | 89.30 (84.98–93.04) | 95.80 (93.41–98.44) | 93.52 (90.10–96.53) |
| | ROI-restricted | 87.26 (82.91–91.20) | 89.21 (84.79–92.87) | 95.56 (92.44–97.66) | 91.74 (88.47–95.29) |
| RF-Deep | ROI features | 95.80 (92.60–98.50) | 93.30 (89.90–96.00) | 100.00 (99.90–100.00) | 100.00 (100.00–100.00) |
| *FPR95* | | | | | |
| MaxSoftmax | All voxels | 38.37 (31.78–49.24) | 49.27 (39.53–59.78) | 23.26 (11.47–34.19) | 34.27 (24.80–49.30) |
| | ROI-restricted | 38.02 (29.50–48.58) | 46.41 (37.41–55.77) | 24.58 (10.41–35.25) | 44.01 (32.34–58.27) |
| MaxLogit | All voxels | 40.01 (24.80–52.52) | 41.59 (29.84–53.96) | 18.05 (10.41–27.37) | 30.87 (16.17–56.87) |
| | ROI-restricted | 36.73 (26.96–50.40) | 42.57 (33.06–55.43) | 20.00 (11.13–28.06) | 35.67 (22.30–52.63) |
| Energy | All voxels | 39.86 (24.80–52.55) | 43.04 (29.84–55.43) | 17.47 (10.41–26.65) | 29.96 (15.83–56.87) |
| | ROI-restricted | 37.06 (26.62–50.74) | 42.25 (33.06–55.05) | 19.99 (11.85–27.72) | 35.16 (20.83–52.63) |
| RF-Deep | ROI features | 15.20 (7.10–23.50) | 25.50 (16.80–36.20) | 0.10 (0.00–1.00) | 0.00 (0.00–0.00) |

Table 12: RF-Deep OOD detection performance across different pretrained backbone encoders. AUROC ($\uparrow$) and FPR95 ($\downarrow$) with 95% bootstrap confidence intervals are reported over 100 matched-seed runs.

| Model | Method | RSNA PE | | MIDRC C19$^-$ | | KiTS | | PancreasCT | |
| | | AUROC | FPR95 (%) | AUROC | FPR95 (%) | AUROC | FPR95 (%) | AUROC | FPR95 (%) |
|---|---|---|---|---|---|---|---|---|---|
| SimMIM | MaxSoftmax | 86.89 | 45.56 | 84.58 | 54.92 | 94.19 | 21.59 | 94.30 | 24.04 |
| | MaxLogit | 89.39 | 44.04 | 87.55 | 45.65 | 95.99 | 15.32 | 96.28 | 14.82 |
| | Energy | 89.70 | 44.64 | 87.75 | 45.24 | 96.15 | 14.92 | 96.39 | 13.84 |
| | RF-Radiomics | 87.20 | 48.90 | 89.80 | 34.90 | 97.00 | 15.80 | 96.60 | 17.90 |
| | MD-Deep | 88.60 | 41.32 | 86.85 | 51.72 | 97.66 | 3.40 | 97.84 | 8.30 |
| | RF-Deep | 93.70 | 17.60 | 93.20 | 26.50 | 99.90 | 0.10 | 99.90 | 0.60 |
| iBOT | MaxSoftmax | 83.56 | 48.25 | 80.97 | 52.10 | 90.45 | 32.22 | 88.98 | 36.23 |
| | MaxLogit | 87.31 | 38.55 | 85.24 | 39.33 | 93.82 | 23.45 | 92.29 | 27.17 |
| | Energy | 87.90 | 38.62 | 85.84 | 39.61 | 94.49 | 22.02 | 92.96 | 25.17 |
| | RF-Radiomics | 89.30 | 50.90 | 90.50 | 29.70 | 96.80 | 14.70 | 96.80 | 15.20 |
| | MD-Deep | 87.83 | 44.87 | 86.12 | 52.89 | 98.63 | 6.62 | 96.16 | 17.27 |
| | RF-Deep | 93.60 | 22.60 | 92.70 | 24.60 | 99.80 | 1.10 | 99.80 | 1.10 |
| SMIT Lite | MaxSoftmax | 77.96 | 56.05 | 74.44 | 60.96 | 80.77 | 48.42 | 87.35 | 41.45 |
| | MaxLogit | 78.52 | 53.32 | 79.03 | 51.95 | 79.46 | 45.97 | 85.42 | 41.33 |
| | Energy | 78.16 | 53.82 | 79.37 | 52.22 | 78.74 | 45.59 | 83.90 | 40.53 |
| | RF-Radiomics | 88.90 | 38.90 | 90.70 | 31.10 | 97.10 | 13.80 | 97.40 | 12.80 |
| | MD-Deep | 85.94 | 38.85 | 78.51 | 56.77 | 93.51 | 25.57 | 92.26 | 25.06 |
| | RF-Deep | 94.10 | 20.40 | 90.40 | 30.50 | 99.60 | 1.80 | 99.30 | 1.80 |
| Swin UNETR | MaxSoftmax | 80.30 | 52.19 | 79.65 | 62.01 | 84.40 | 46.38 | 88.97 | 46.98 |
| | MaxLogit | 83.88 | 42.62 | 83.85 | 52.12 | 88.99 | 34.86 | 91.57 | 35.45 |
| | Energy | 84.16 | 40.77 | 84.30 | 50.44 | 89.41 | 31.86 | 92.17 | 33.91 |
| | RF-Radiomics | 87.70 | 37.40 | 89.20 | 33.40 | 98.10 | 9.80 | 98.10 | 13.70 |
| | MD-Deep | 77.88 | 62.47 | 74.14 | 64.61 | 92.61 | 25.50 | 89.54 | 29.67 |
| | RF-Deep | 89.60 | 34.10 | 88.90 | 37.50 | 99.70 | 2.40 | 99.60 | 2.90 |
| Swin UNETR$^\dagger$ | MaxSoftmax | 75.92 | 53.25 | 77.83 | 55.91 | 78.89 | 53.17 | 93.12 | 23.96 |
| | MaxLogit | 77.87 | 43.65 | 80.41 | 49.93 | 83.33 | 39.30 | 92.37 | 18.44 |
| | Energy | 78.02 | 42.46 | 80.78 | 49.48 | 83.87 | 38.24 | 92.17 | 18.98 |
| | MD-Deep | 74.03 | 68.07 | 70.50 | 71.02 | 84.56 | 49.44 | 79.45 | 63.34 |
| | RF-Radiomics | 86.50 | 40.00 | 89.00 | 37.20 | 97.40 | 11.30 | 97.80 | 11.70 |
| | RF-Deep | 90.40 | 41.50 | 88.60 | 35.80 | 99.40 | 2.60 | 98.70 | 7.10 |

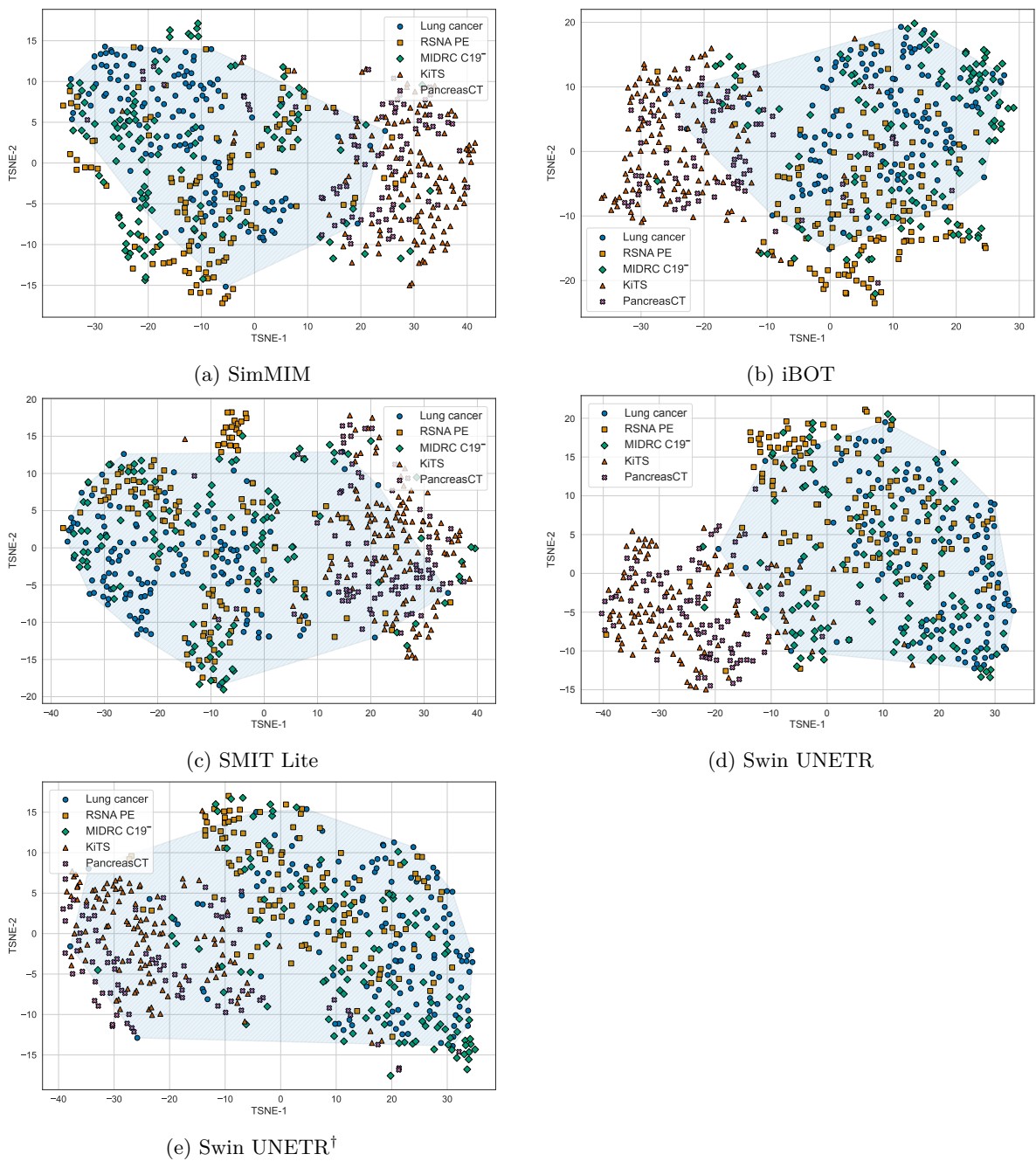

Figure 15: t-SNE projected embeddings showing dataset-wise separability of (a) deep features and (b) radiomics features. Results from one representative split (of 100), combining all datasets in a single visualization, are shown for brevity. The shaded blue regions indicate the convex hull of the ID dataset.

