# OpenReview forum: "Tumor-anchored deep feature random forests for out-of-distribution detection in lung cancer segmentation"
_TMLR — Accepted by TMLR_

### Review · Reviewer_uU1P · 2026-03-05

**Summary Of Contributions:**

This paper proposes RF‑Deep, a post‑hoc, plug‑and‑play scan‑level OOD detector for 3D CT lung tumor segmentation. It trains a random forest with limited outlier exposure on hierarchical encoder features extracted from a frozen segmentation model, using multiple ROIs anchored to predicted tumor regions to make detection task‑relevant. Evaluation on 2,056 CT scans spanning ID lung cancer, near‑OOD chest CT (PE, COVID‑19 negative), and far‑OOD abdominal CT (kidney cancer, healthy pancreas) shows improved AUROC/FPR95 over baselines.

**Audience:**

Yes

**Audience Explanation:**

OOD detection for medical segmentation is practically important for the application of AI in healthcare.

**Claims And Evidence:**

No

**Claims Explanation:**

While the paper reports strong AUROC/FPR95 with bootstrap confidence intervals and provides a clear method description, several central claims require additional controls/clarifications to be fully convincing.
1. Near‑OOD confounding by scanner/protocol/dataset: The near‑OOD datasets (RSNA PE; MIDRC COVID‑19 negative) are different public datasets than the NSCLC datasets used for segmentation training/testing, so "ID vs near‑OOD" may inadvertently include dataset/scanner/protocol differences. The paper itself highlights imaging variation factors (scanner manufacturer, contrast status, reconstruction kernels) and states fine‑tuning was performed under a specific scanner/protocol regime, making it plausible that the RF learns acquisition signatures rather than pathology differences unless explicitly controlled.
2. Far‑OOD may be trivially easy: Near‑perfect performance on abdominal datasets (kidney cancer; healthy pancreas) could reflect coarse anatomical/FOV differences rather than task‑relevant OOD detection; this should be controlled for or claims should be narrowed accordingly.
3. Baseline fairness: Logit/energy baselines are computed by mean‑aggregating tumor‑class predictions across all voxels; stronger scan‑level pooling or ROI‑restricted variants are needed to ensure the comparative evidence supports "substantial outperformance."

**Requested Changes:**

1. Control for scanner/protocol confounding in near‑OOD. Add leave‑one‑scanner/site/protocol out tests within NSCLC: hold out one scanner/protocol stratum as ID test and report how often RF flags it as OOD. Where possible, also stratify near‑OOD by scanner/protocol or match subsets.
2. Add controls to reduce trivial anatomy/FOV cues (e.g., crop/mask/normalize) or narrow claims so far‑OOD is presented as a sanity check.
3. Make outlier‑exposure protocol explicit. Clearly list which OOD datasets are used for OE vs held out for each experiment (and any tuning).
4. Strengthen baseline aggregation. Add stronger scan‑level uncertainty baselines (max/percentile pooling; ROI‑restricted pooling) instead of voxel‑mean only.
5. Quantify anchoring failure modes. Report performance for "no/tiny predicted tumor" cases and stratify by predicted component size/count/confidence.

---

> ### Author Response · Authors · 2026-03-09
>
> Thank you for your review and positive feedback. We appreciate your time and thoughtful analysis and will revert back shortly.

---

> ### Author Response · Authors · 2026-03-27
> **Response to reviewer (1 of 6)**
>
> We thank the reviewer for the thoughtful assessment. In response, we conducted leave-one-factor-out analyses for scanner, contrast, and reconstruction kernel variation, strengthened baseline aggregation protocols, expanded evaluation on strictly unseen OOD datasets, and clarified anchoring failure modes and deployment assumptions. We believe these additions substantially strengthen the paper.
>
> **Concern 1: Near-OOD confounding by scanner/protocol/dataset**
>
> We conducted three experiments to directly test whether RF-Deep learns scanner/protocol signatures rather than pathological differences by assessing OOD detection on the ID datasets. OOD classifiers were trained to different ID (lung cancer) versus other diseases (e.g. RSNA PE, MIDRC COVID-19 negative, etc). Testing was performed to detect shortcuts due to scanner/protocol bias by evaluating *dataset-specific* RF-Deep classifiers and aggregating the average OOD detections, and to detect shortcuts due to dataset-specific bias by using the dataset-specific RF-Deep models as an *ensemble* (decision-level) classifier. Three different experiments were performed to assess dependence on scanner manufacturer, image contrast, and reconstruction kernel to produce CTs.
>
> ***Experiment 1: Evaluating scanner bias using Leave-one-scanner-out OOD classifier training.***
>
> | Held-out scanner | Method | False OOD rate (%) | OOD detection AUROC |
> |---|---|---|---|
> | GE | Dataset-specific | 9.07 (6.19--12.62) | 97.95 (96.83--98.82) |
> | GE | Ensemble | 1.31 (0.00--4.31) | 95.67 (93.95--96.97) |
> | Siemens | Dataset-specific | 10.15 (4.35--18.48) | 97.98 (96.58--98.94) |
> | Siemens | Ensemble | 2.52 (0.00--8.70) | 96.19 (94.03--97.63) |
>
> The NSCLC Radiogenomics test set ($N{=}140$) includes scans from multiple scanner manufacturers (GE: $N{=}105$, Siemens: $N{=}23$). We trained RF-Deep using a fixed training size of 20 ID scans drawn from the non-held-out scanner group plus OOD data, and measured the false OOD rate on the held-out scanner. Under the ensemble strategy, false OOD rates remained low (GE: 1.3%, Siemens: 2.5%), indicating that RF-Deep does not simply flag scanner manufacturer differences as OOD. OOD detection AUROC remained strong at 95.7 and 96.2, suggesting that the detector still functions effectively even when the ID training data is restricted to a single scanner.
>
> ***Experiment 2: Evaluating image contrast bias using Leave-one-contrast-status-out OOD classifier training.***
>
> | Held-out protocol | Method | False OOD rate (%) | OOD detection AUROC |
> |---|---|---|---|
> | Contrast | Dataset-specific | 8.75 (5.57--14.72) | 98.14 (95.47--99.03) |
> | Contrast | Ensemble | 0.87 (0.00--3.53) | 96.31 (92.31--97.96) |
> | Non-Contrast | Dataset-specific | 14.70 (9.72--20.61) | 96.69 (95.17--97.81) |
> | Non-Contrast | Ensemble | 8.39 (0.00--18.52) | 93.71 (91.45--95.57) |
>
> Contrast-enhanced scans showed very low false OOD rates under the ensemble strategy (0.9%), while non-contrast scans showed a moderate rate (8.4%). This asymmetry is consistent with the training data composition, where contrast-enhanced scans are more prevalent. OOD detection AUROC remained above 93 in all configurations, indicating that contrast status does not prevent effective OOD detection.
>
> ***Experiment 3: Evaluating image reconstruction bias using Leave-one-reconstruction-kernel-out.***
>
> Using the kernel categories defined in Section 5.1 (Recon 1/2/3), we held out each category in turn:
>
> | Held-out kernel | Method | False OOD rate (%) | OOD detection AUROC |
> |---|---|---|---|
> | Recon 1 | Dataset-specific | 43.74 (34.56--54.83) | 91.60 (87.55--94.36) |
> | Recon 1 | Ensemble | 42.68 (14.71--70.59) | 85.87 (79.67--90.66) |
> | Recon 2 | Dataset-specific | 3.74 (1.32--11.18) | 99.13 (98.50--99.58) |
> | Recon 2 | Ensemble | 1.16 (0.00--2.63) | 98.32 (97.18--99.23) |
> | Recon 3 | Dataset-specific | 3.66 (0.78--11.35) | 98.47 (96.29--99.73) |
> | Recon 3 | Ensemble | 1.45 (0.00--6.25) | 96.33 (91.89--98.95) |
>
> Recon 2 and Recon 3 kernels showed low false OOD rates under the ensemble strategy (1.16% and 1.45%, respectively), suggesting that RF-Deep does not generally confound reconstruction kernel differences with OOD. The increased rate in Recon 1 kernels (42.7--43.7%) likely reflects cumulative stacked shifts (GE manufacturer plus non-contrast protocol), highlighting sensitivity under highly specific conditions rather than failure to generalize kernel signatures.
>
> Further, if kernel signatures were the primary driver, Recon 2 and 3 would show similar or worse degradations, which was not observed. The leave-one-factor-out design is maximally adversarial (for the purpose of this ablation study), completely excluding the held-out subgroup from the RF's 20 ID training examples (or more, as required); in deployment, including even a small number of representative scans from each kernel category would mitigate this effect. (contd.)

---

> > ### Author Response · Authors · 2026-03-27
> > **Response to reviewer (2 of 6)**
> >
> > (contd.) The leave-one-factor-out experiments within the ID cohort evaluated robustness to fairly comprehensive known set of confounders for CT-based medical image analysis. The consistent SHAP feature rankings across all six OOD cohorts (including the two blinded validation datasets from different institutions) provide indirect evidence that RF-Deep's decisions are driven by pathological rather than acquisition-related signals. Nevertheless, we concede that these experiments cannot fully decouple the interaction of all these factors and pathology across all datasets because the near-OOD cohorts were acquired independently and by different institutions with different clinical and image acquisition protocols. However, this evaluation also presents a more realistic scenario as models are trained in isolation and then the OOD cohorts are presented in the clinical settings. Hence, this limitation is shared by all evaluated methods and does not affect relative comparisons.
> >
> > This result has been updated in the revised paper (Section 5.4.3).
> >
> > **Concern 2: Far-OOD may be trivially easy.**
> >
> > We agree that far-OOD detection is inherently easier due to anatomical differences. Nevertheless, this experiment is important to evaluate for sanity check. Of note, prior works on medical image analysis always include far-OOD analysis and we felt it is important to include this to contextualize our work in relation to others. In keeping with reviewer's suggestions,
> >
> > - The narrative now leads with the clinically challenging near-OOD results, where RF-Deep achieves the largest improvement over baselines.
> > - Far-OOD results are retained as per the original draft, but framed as validation that RF-Deep reliably catches routing errors as well, a common deployment failure mode motivated in the Introduction of the paper (PACS routing of non-thoracic scans to lung-specific pipelines).
> > - We have, throughout the paper, acknowledged that far-OOD performance at times benefits from anatomical differences.
> >
> > We also note that the tumor-anchored ROI extraction ($128^3$ crops centered on predicted lesions) limits the spatial context available to the RF to local regions around detections, rather than global FOV information. Our SHAP analysis (Figure 6c--d) shows that far-OOD detection relies on features from Stages 1--2 (high-resolution, tissue-level features) alongside later stages, and hence, we position far-OOD evaluation primarily as a sanity check for routing errors rather than as the central contribution.
> >
> > To further demonstrate that RF-Deep's value lies in the challenging near-OOD regime, we also evaluated on two blinded validation datasets, where the classifier achieves strong performance without any exposure to those categories during training. Their results are reported in the forthcoming sections.
> >
> > **Concern 3: Make outlier‑exposure protocol explicit.**
> >
> > We apologize for the insufficient clarity. Here is a comprehensive data split table that mentions the exact datasets for every experiment protocol:
> >
> > | Strategy | OE Train | OOD Test |
> > |---|---|---|
> > | Dataset-specific | Target OOD dataset only | Target OOD dataset (70%) |
> > | Ensemble | Same as dataset-specific (4x) | All OOD (predictions averaged) |
> > | LODO | 3 of 4 OOD datasets | Held-out 4th dataset (70%) |
> >
> > For RF-Deep (and RF-Radiomics and MD-Deep), 100 independent runs were performed with unique random seeds, each generating a fresh 30%/70% train/test split and sampling $N{=}20$ ID and $N{=}20$ OE scans for training; 95% confidence intervals reflect the percentile range across these 100 runs. For logit-based baselines (MaxSoftmax, MaxLogit, Energy), 95% confidence intervals were computed by bootstrap resampling of scan-level scores over 100 iterations. This has now been made clear in the paper (Section 3.3).

---

> > > ### Author Response · Authors · 2026-03-27
> > > **Response to reviewer (3 of 6)**
> > >
> > > **Concern 4: Strengthen baseline aggregation.**
> > >
> > > We conducted two sets of experiments comparing different methods under identical tumor-anchored ROIs and matched seeds to ensure fair comparison.
> > >
> > > We conducted two sets of experiments comparing different methods under identical tumor-anchored ROIs and matched seeds to ensure fair comparison.
> > >
> > > ***Experiment 1: ROI-restricted logit baselines.***
> > >
> > > We re-computed MaxSoftmax, MaxLogit, and Energy using only voxels within the same $n{=}4$ tumor-anchored $128^3$ ROIs used by RF-Deep, averaging scores across ROIs at the scan level to exactly mirror RF-Deep's spatial protocol:
> > >
> > > | Method | Aggregation | RSNA PE | MIDRC C19$^-$ | KiTS | PancreasCT |
> > > |---|---|---|---|---|---|
> > > | *AUROC* | | | | | |
> > > | MaxSoftmax | All voxels | 88.61 (84.62--92.36) | 86.57 (81.79--90.14) | 95.67 (93.41--98.06) | 94.61 (92.61--97.49) |
> > > | MaxSoftmax | ROI-restricted | 86.43 (82.05--90.33) | 87.22 (81.91--91.42) | 95.57 (93.07--97.63) | 89.28 (85.66--93.43) |
> > > | MaxLogit | All voxels | 88.77 (85.09--92.43) | 89.31 (85.05--93.06) | 95.89 (93.58--98.44) | 93.53 (90.17--96.59) |
> > > | MaxLogit | ROI-restricted | 87.38 (83.06--91.14) | 89.18 (84.72--92.75) | 95.79 (92.92--97.73) | 91.63 (88.32--95.23) |
> > > | Energy | All voxels | 88.59 (84.85--92.24) | 89.30 (84.98--93.04) | 95.80 (93.41--98.44) | 93.52 (90.10--96.53) |
> > > | Energy | ROI-restricted | 87.26 (82.91--91.20) | 89.21 (84.79--92.87) | 95.56 (92.44--97.66) | 91.74 (88.47--95.29) |
> > > | RF-Deep | ROI features | 95.80 (92.60--98.50) | 93.30 (89.90--96.00) | 100.00 (99.90--100.00) | 100.00 (100.00--100.00) |
> > > | *FPR95* | | | | | |
> > > | MaxSoftmax | All voxels | 38.37 (31.78--49.24) | 49.27 (39.53--59.78) | 23.26 (11.47--34.19) | 34.27 (24.80--49.30) |
> > > | MaxSoftmax | ROI-restricted | 38.02 (29.50--48.58) | 46.41 (37.41--55.77) | 24.58 (10.41--35.25) | 44.01 (32.34--58.27) |
> > > | MaxLogit | All voxels | 40.01 (24.80--52.52) | 41.59 (29.84--53.96) | 18.05 (10.41--27.37) | 30.87 (16.17--56.87) |
> > > | MaxLogit | ROI-restricted | 36.73 (26.96--50.40) | 42.57 (33.06--55.43) | 20.00 (11.13--28.06) | 35.67 (22.30--52.63) |
> > > | Energy | All voxels | 39.86 (24.80--52.55) | 43.04 (29.84--55.43) | 17.47 (10.41--26.65) | 29.96 (15.83--56.87) |
> > > | Energy | ROI-restricted | 37.06 (26.62--50.74) | 42.25 (33.06--55.05) | 19.99 (11.85--27.72) | 35.16 (20.83--52.63) |
> > > | RF-Deep | ROI features | 15.20 (7.10--23.50) | 25.50 (16.80--36.20) | 0.10 (0.00--1.00) | 0.00 (0.00--0.00) |
> > >
> > > Restricting voxel-wise uncertainty to the same tumor-anchored ROIs used by RF-Deep did not improve the logit-based baselines. Across the primary benchmark cohorts, ROI restriction produced only negligible changes in AUROC and in several cases slightly reduced performance relative to the original all-voxel aggregation. The same pattern was observed on the two blinded validation datasets, where ROI restriction yielded only modest gains for MaxSoftmax on Breast Cancer CT, and otherwise did not improve OOD detection. In contrast, RF-Deep maintained a clear advantage across all cohorts, particularly on the near-OOD lung datasets. These results indicate that RF-Deep's superiority is not explained by spatial focus alone; rather, its advantage is consistent with the richer hierarchical feature representation captured within the ROIs. This result has been updated in the revised paper (Table 11).
> > >
> > > ***Experiment 2: Scan-level pooling.***
> > >
> > > For voxel-wise uncertainty baselines, scan-level aggregation is a legitimate design choice that could materially affect OOD performance. We therefore re-evaluated MaxLogit under four foreground-restricted pooling rules: mean (our original implementation), max, 95th percentile, and median.
> > >
> > > | Method | Pooling | RSNA PE | MIDRC C19$^-$ | KiTS | PancreasCT |
> > > |---|---|---|---|---|---|
> > > | *AUROC* | | | | | |
> > > | MaxLogit | Mean | 88.77 (85.09--92.43) | 89.31 (85.05--93.06) | 95.89 (93.58--98.44) | 93.53 (90.17--96.59) |
> > > | MaxLogit | Max | 82.59 (78.58--86.76) | 85.30 (79.35--90.32) | 95.00 (91.87--97.19) | 94.51 (89.55--98.19) |
> > > | MaxLogit | 95th percentile | 87.82 (83.92--91.09) | 86.67 (81.21--90.89) | 95.87 (93.37--97.77) | 93.35 (88.09--96.99) |
> > > | MaxLogit | Median | 87.00 (83.00--90.94) | 85.14 (79.87--89.48) | 95.53 (92.83--97.65) | 89.56 (84.41--94.07) |
> > > | RF-Deep | - | 95.80 (92.60--98.50) | 93.30 (89.90--96.00) | 100.00 (99.90--100.00) | 100.00 (100.00--100.00) |
> > > | *FPR95* | | | | | |
> > > | MaxLogit | Mean | 40.01 (24.80--52.52) | 41.59 (29.84--53.96) | 18.05 (10.41--27.37) | 30.87 (16.17--56.87) |
> > > | MaxLogit | Max | 74.60 (55.04--89.51) | 58.07 (28.88--84.06) | 23.38 (11.56--41.79) | 28.35 (6.87--96.38) |
> > > | MaxLogit | 95th percentile | 42.34 (28.61--62.70) | 42.64 (27.16--66.67) | 19.57 (11.21--31.92) | 34.62 (10.49--84.09) |
> > > | MaxLogit | Median | 37.93 (28.26--50.38) | 41.46 (30.43--51.11) | 20.51 (13.04--29.37) | 44.97 (22.46--84.06) |
> > > | RF-Deep | - | 15.20 (7.10--23.50) | 25.50 (16.80--36.20) | 0.10 (0.00--1.00) | 0.00 (0.00--0.00) |
> > >
> > > (contd.)

---

> ### Author Response · Authors · 2026-03-27
> **Response to reviewer (4 of 6)**
>
> (contd.)
>
> Across the primary benchmark datasets, no alternative pooling rule closed the gap to RF-Deep. Mean pooling remained the strongest or near-strongest choice on MIDRC C19$^-$ and KiTS, while max pooling improved performance only for PancreasCT and substantially worsened results on the near-OOD lung datasets, consistent with oversensitivity to isolated extreme voxels. The 95th-percentile rule provided the most balanced alternative overall, but still remained clearly below RF-Deep on both AUROC and FPR95. For example, on RSNA PE the best MaxLogit variant achieved an AUROC of 88.77, compared with 95.80 for RF-Deep, and on MIDRC C19$^-$, the best MaxLogit result was 89.31 versus 93.30 for RF-Deep.
>
> MaxSoftmax showed the same overall behavior as MaxLogit, wherein, changing the scan-level pooling did not confer any performance improvement in comparison to RF-Deep, particularly on the near-OOD lung datasets. This indicates that RF-Deep's advantage is not attributable to a favorable aggregation choice for voxel-wise confidence scores.
>
> We did not emphasize analogous pooling ablations for Energy because, unlike MaxLogit and MaxSoftmax, Energy is already a transformed summary of the full class-logit vector at each voxel; applying additional extreme-value pooling over foreground voxels is therefore less directly interpretable. Under its standard scan-level aggregation, Energy also remained below RF-Deep on the benchmark datasets.
>
> **Concern 5: Quantifying failure modes and beyond**
>
> This is an excellent question. In general, we posit that for safe use, AI models used in clinical situations should always be overseen by an expert. The scenario the reviewer raises can occur under two conditions: (a) in the ID dataset, when real tumors are missed, which is a failure of the model but not the OOD detector. (b) in the OOD dataset, where no detection is a desired feature and would eliminate the need for an OOD detector altogether. There is also a third case, where tumors are falsely detected in the ID dataset that cannot be handled by our approach, which is why an expert supervision is always needed. The scenario that is the most problematic is due to additional resources required by the expert to reject false detections in the OOD cases, and would diminish the usability of any model, which is what we focused on in this work. As reviewer pointed out, it is important to thoroughly investigate the failure modes in this case, including performance when the segmented tumors are small.
>
> ***Analysis 1: Stratification by predicted tumor volume.***
>
> We stratified OOD scans by predicted foreground volume using quartile boundaries derived from the ID test set (NSCLC) within each run and applied those absolute thresholds consistently across all OOD datasets. Scans without prediction were excluded from quartile assignment. Across runs, the mean ID-derived Q1/Q2, Q2/Q3, and Q3/Q4 boundaries were 4.32 cc (range, 3.25--5.15), 11.35 cc (range, 8.51--12.90), and 28.75 cc (range, 21.77--32.75), respectively.
>
> To assess whether detection performance was volume-dependent, we report the Spearman correlation between predicted foreground volume and mean OOD probability per scan. Both detectors maintained strong performance across all volume strata. For the far-OOD datasets, performance was near-perfect and volume-independent ($|\rho|$ $\leq$ 0.23, $p \geq 0.185$), consistent with detection being driven by gross anatomical differences rather than lesion size.
>
> For the near-OOD datasets, a moderate positive correlation was observed for RSNA PE ($\rho$=0.48 and 0.39 for dataset-specific and ensemble, both $p < 0.001$), indicating that larger detections were assigned higher OOD probability and that smaller detections yielded less distinctive encoder representations relative to the ID distribution, reducing their separability. No such relationship was found for MIDRC C19$^-$ under the ensemble detector ($\rho$=0.16, $p$=0.214), suggesting that detection in that cohort is not primarily size-driven. Additionally, the same volume-dependent pattern was observed in both blinded validation datasets ($\rho$ = 0.36 and 0.56, both $p < 0.001$), with results identical across detectors, providing encouraging evidence that the relationship generalizes beyond the OE training distribution.
>
> | Dataset | Dataset-specific $\rho$ | Dataset-specific $p$ | Ensemble $\rho$ | Ensemble $p$ |
> |---|---|---|---|---|
> | RSNA PE | 0.48 | <0.001 | 0.39 | <0.001 |
> | MIDRC C19$^-$ | 0.26 | 0.038 | 0.16 | 0.214 |
> | KiTS23 | 0.02 | 0.894 | 0.23 | 0.037 |
> | PancreasCT | -0.16 | 0.271 | -0.19 | 0.185 |
> | MIDRC C19$^+$ | 0.36 | <0.001 | 0.36 | <0.001 |
> | Breast cancer CT | 0.56 | <0.001 | 0.56 | <0.001 |
>
> This result has been updated in the revised paper (Section 5.2.3, Figure 5).
> (contd.)

---

> ### Author Response · Authors · 2026-03-27
> **Response to reviewer (5 of 6)**
>
> (contd.)
>
> ***Analysis 2: Tiny predicted tumor OOD detection performance.***
>
> To specifically examine the effect of lesion size (tiny tumors), we further stratified OOD scans by predicted foreground volume and evaluated detection performance on the smallest volume bin based on ID statistics (Q1: 0--4.32 cc), reporting both mean OOD probability and the fraction of scans with probability $<$ 0.5. For far-OOD datasets (KiTS, PancreasCT), detection remained uniformly robust: no small-volume scans were misclassified (0/33 and 0/27, respectively), with high mean OOD probabilities (0.82--0.93). This particular scenario indicates that detection in these scans is primarily driven by anatomical mismatch rather than lesion size.
>
> In contrast, near-OOD datasets revealed a clear method-dependent behavior. The dataset-specific detector maintained strong performance even for small lesions (RSNA-PE: 6%; MIDRC-C19$^-$: 17%; mean probability $\approx$ 0.67), consistent with its access to target-domain characteristics at inference. The ensemble detector, however, exhibited substantially higher miss rates (RSNA-PE: 69%; MIDRC-C19$^-$: 54%; mean probability $\approx$ 0.45), with predictions concentrated near the decision boundary rather than being confidently incorrect. This pattern suggests that the challenge is not lesion size, but the ambiguity of small, near-OOD appearances under cross-distribution generalization. The ensemble detector, designed to generalize across different domains, produces more conservative scores in such cases, as supported by the positive correlation between predicted volume and OOD probability (Spearman $\rho = 0.39$, $p < 0.001$ for RSNA-PE). Notably, for the two unseen datasets, both detectors performed identically (MIDRC-C19$^+$: 64%; breast cancer CT: 69%), indicating that the observed limitation is not specific to the ensemble design but reflects an inherent difficulty in detecting small, ambiguous near-OOD tumors without prior exposure.
>
> ***Analysis 3: Stratification by connected component count.***
>
> | Dataset | 0 (no pred.) | 1 | 2--5 | >5 | AUROC Dataset-specific | AUROC Ensemble |
> |---|---|---|---|---|---|---|
> | RSNA PE | 10.1 | 21.8 | 44.5 | 23.5 | 95.80 (92.60--98.50) | 93.70 (91.70--96.50) |
> | MIDRC C19$^-$ | 32.3 | 22.9 | 31.3 | 13.5 | 93.30 (89.90--96.00) | 92.80 (89.80--95.60) |
> | KiTS | 32.2 | 23.7 | 33.1 | 11.0 | 100.00 (99.90--100.00) | 100.00 (99.80--100.00) |
> | PancreasCT | 34.2 | 34.2 | 28.9 | 2.60 | 100.00 (100.00--100.00) | 100.00 (100.00--100.00) |
> | MIDRC C19$^+$ | 9.10 | 20.0 | 43.6 | 27.3 | 90.92 (79.31--97.13) | 94.99 (91.69--96.97) |
> | Breast cancer CT | 28.8 | 30.3 | 40.9 | 0.00 | 94.75 (87.34--99.26) | 96.96 (93.38--99.25) |
> | ID (NSCLC) | 0.70 | 25.2 | 66.2 | 7.90 | --- | --- |
>
> On scans where anchoring succeeds ($\geq$1 kept component), both detectors maintain strong AUROC across all datasets, including the two blinded validation datasets. No-prediction rates were substantially higher for far-OOD datasets (KiTS: 32.2%, PancreasCT: 34.2%) and MIDRC C19$^-$ (32.3%) than for ID scans (0.7%), indicating that complete anchor failure itself carries useful OOD information.
>
> (contd.)

---

> > ### Author Response · Authors · 2026-03-27
> > **Response to reviewer (6 of 6)**
> >
> > (contd.)
> >
> > ***Analysis 4: Enhanced pipeline and ID specificity.***
> >
> > To quantify the utility of the no-prediction signal, we evaluated a simple enhanced policy: assign OOD score $= 1.0$ to scans with absent prediction, while retaining the RF-Deep probability otherwise. Under this policy, the ensemble AUROC matched or exceeded the standard AUROC on the near-OOD datasets, while introducing only minor degradation on far-OOD datasets where performance was already near-perfect. This concern was also raised by Reviewers q5h1 (Concern 3) and QbSa (Concern 3).
> >
> > | Method | RSNA PE | MIDRC C19$^-$ | KiTS | PancreasCT |
> > |---|---|---|---|---|
> > | *Dataset-Specific* | | | | |
> > | *AUROC* | | | | |
> > | Standard | 95.80 (92.60--98.50) | 93.30 (89.90--96.00) | 100.00 (99.90--100.00) | 100.00 (100.00--100.00) |
> > | Enhanced | 96.14 (92.80--98.29) | 95.43 (92.53--97.34) | 99.42 (99.07--100.00) | 99.44 (99.12--100.00) |
> > | *FPR95 (%)* | | | | |
> > | Standard | 15.20 (7.10--23.50) | 25.50 (16.80--36.20) | 0.10 (0.00--1.00) | 0.00 (0.00--0.00) |
> > | Enhanced | 15.43 (6.61--24.49) | 19.96 (12.73--29.11) | 0.68 (0.00--1.02) | 0.67 (0.00--1.02) |
> > | *Ensemble* | | | | |
> > | *AUROC* | | | | |
> > | Standard | 93.70 (91.70--96.50) | 92.80 (89.80--95.60) | 100.00 (99.80--100.00) | 100.00 (100.00--100.00) |
> > | Enhanced | 94.17 (91.39--97.16) | 94.97 (93.18--96.81) | 99.41 (98.92--100.00) | 99.44 (99.12--100.00) |
> > | *FPR95 (%)* | | | | |
> > | Standard | 19.90 (11.20--29.10) | 32.70 (20.40--54.10) | 0.00 (0.00--0.00) | 0.00 (0.00--0.00) |
> > | Enhanced | 20.39 (11.71--29.59) | 26.12 (14.77--41.40) | 0.67 (0.00--1.02) | 0.67 (0.00--1.02) |
> >
> > The enhanced policy introduced only 2.73 mean ID false positives per run (out of 98 ID test scans; approximately 2.8%), indicating that the no-prediction signal does not materially degrade in-distribution specificity. This behavior was consistent across all six OOD datasets, including the two blinded validation datasets. Higher no-prediction rates in far-OOD datasets (KiTS: 32.2%, PancreasCT: 34.2%) and MIDRC C19$^-$ (32.3%) therefore appear to function as a complementary and anatomically interpretable OOD signal, rather than merely reflecting detector failure.
> >
> > However, as its severity cannot be fully characterized across all deployment scenarios with the currently available in-distribution datasets, we only consider RF-Deep if predictions are present; and flag others for primary manual review. Future work may explore developing a two-stage approach that leverages the no-prediction signal (as well as other surrounding factors) as complementary, anatomically interpretable ID/OOD indicator. This result has been updated in the revised paper (Section 5.2.3).
> >
> > We hope these additions address the reviewer's concerns and strongly believe the revised manuscript now presents a clearer and more carefully scoped evaluation of RF-Deep.

---

### Review · Reviewer_q5h1 · 2026-03-08

**Summary Of Contributions:**

This paper introduces RF-Deep, a lightweight, post-hoc out-of-distribution (OOD) detection framework for medical image segmentation. Its core innovation is to extract deep features from a frozen, SSL-pretrained encoder anchored to the model's predicted segmentation masks. These tumor-anchored features are used to train a random forest classifier with outlier exposure. The framework is evaluated on lung cancer CT segmentation, demonstrating significant improvements over logit-based and radiomics baselines. The work provides a practical, architecture-agnostic plug-in to enhance the reliability of deployed segmentation models.

**Audience:**

Yes

**Audience Explanation:**

This paper addresses the models' failure to recognize out-of-distribution inputs, which is a critical bottleneck in clinical AI deployment. While random forests and deep features are not new, their combination with tumor-anchored ROIs and systematic analysis across pretraining strategies provides genuine insight. In addition, this proposed method requires no retraining of the base model, which shows its lightweight and practicality.

**Broader Impact Concerns:**

The authors have provided a thoughtful and well-structured Broader Impact Statement. They correctly identify the positive potential of their work in improving patient safety by preventing silent failures. They also responsibly acknowledge the limitations, stating that "no OOD detection method is perfect." The paper's thorough evaluation and acknowledgment of this limitation mitigate this concern. The work is a clear step forward in responsible AI development for healthcare.

**Claims And Evidence:**

Yes

**Claims Explanation:**

1.	The paper is well-organized, and the writing is generally clear and concise. The extensive experiments are conduced to prove their claims.
2.	The proposed RF-Deep framework is clever in its simplicity. Anchoring feature extraction to the predicted segmentation mask focuses the OOD detector on task-relevant regions, and using a lightweight random forest on top of a frozen foundation model makes it a practical, plug-and-play solution.
3.	The paper provides insightful analysis by SHAP analysis and t-SNE visualizations, which clearly explain why deep features outperform radiomics and why RF surpasses linear/MLP classifiers.

**Requested Changes:**

1.	No inference time or model parameters comparisons to substantiate "lightweight" claims.
2.	No ablation studies about the hyperparameters in random forests.
3.	The paper notes that in 10% of OOD scans, the segmentation model produces no foreground predictions. Please briefly discuss how this could be integrated into the current workflow or if it represents a fundamental limitation.
4.	This method requires tumor predictions, curated OOD examples, and hyperparameter tuning, which is more involved than truly plug-and-play methods like MaxSoftmax. Is there an overstatement in the "Plug-and-Play" claim?

---

> ### Author Response · Authors · 2026-03-09
>
> Thank you for your review and positive feedback. We appreciate your time and thoughtful analysis and will revert back shortly.

---

> ### Author Response · Authors · 2026-03-27
> **Response to reviewer (1 of 2)**
>
> We thank the reviewer for the careful reading, the positive assessment of our contribution, and the constructive suggestions. We address each requested change below with new experiments and manuscript revision, specifically addressing runtime measurements, systematic RF hyperparameter ablations, handling of no-prediction scans, and refining the deployment framing of RF-Deep as a lightweight post-hoc detector.
>
> **Concern 1: Inference time and model parameters**
>
> We measured inference cost for both backbones for the shared segmentation pass, logit-based methods, and RF-Deep. All timing measurements were performed on a system with a single NVIDIA A40 GPU (46 GB memory; driver 535.54.03) and dual Intel Xeon Platinum 8358 CPUs (32 cores per socket, 2 threads per core).
>
> SMIT contains 66.50M parameters; Swin UNETR contains 62.19M total parameters. Despite fewer parameters, Swin UNETR is slower at inference because its smaller ROI size ($96^3$ vs.\ $128^3$) requires more sliding-window passes per scan.
>
> | Method / Stage | SMIT (s) | Swin UNETR (s) |
> |----------------|----------|----------------|
> | Segmentation inference (shared) | 18.39 | 29.79 |
> | Logit post-processing (MaxSoftmax / MaxLogit / Energy) | 0.50 | 0.50 |
> | RF-Deep feature extraction (4 ROIs) | 0.23 | 0.12 |
> | RF-Deep classifier, dataset-specific (1 RF, CPU) | 0.08 | 0.08 |
> | RF-Deep classifier, ensemble (4 RFs, CPU) | 0.31 | 0.40 |
> | --- | --- | --- |
> | RF-Radiomics overhead ($\\sim$) | 20--120 | 20--120 |
> | MD-Deep overhead, dataset-specific ($\\sim$) | 0.28 | 0.20 |
> | RF-Deep overhead, dataset-specific | 0.31 | 0.20 |
> | RF-Deep overhead, ensemble | 0.51 | 0.51 |
>
> Segmentation dominates runtime in all configurations. RF-Deep adds only 0.31s (SMIT, dataset-specific) to 0.51s (SMIT, ensemble) beyond segmentation, corresponding to overheads of 1.7\% and 2.8\%, respectively. For SwinUNETR, the corresponding figures are 0.20s and 0.51s (0.7\% and 1.7\%). Logit-based methods (MaxSoftmax, MaxLogit, Energy) add approximately 0.50s post-segmentation, independent of backbone.
>
> RF-Radiomics incurs 20--120s of additional radiomics extraction per scan, after which classifier inference is negligible. MD-Deep shares the same frozen encoder features as RF-Deep and appends a single lightweight detector; its per-scan overhead is therefore comparable to the dataset-specific RF-Deep setting. The RF classifier runs entirely on CPU and introduces no additional trainable parameters to the segmentation backbone, keeping the deployment footprint minimal.
>
> These results confirm that segmentation inference dominates total runtime, while RF-Deep contributes only negligible overhead ($\leq$ 2.8\%) as it reuses frozen backbone features post fine-tuning and introduces only a shallow classifier. Its computational cost is comparable to simple logit-based scoring and substantially lower than radiomics-based alternatives. Furthermore, RF-Deep does not increase backbone parameter count and performs classification on CPU, confirming that it is both computationally and parametrically lightweight. This result has been updated in the revised paper (Section 5.4.4).
>
> **Concern 2: Random forest hyperparameter ablation**
>
> We conducted systematic RF hyperparameter ablations for RF-Deep on the RSNA PE near-OOD benchmark across 100 runs (95% percentile CI), evaluating both the dataset-specific RF and the ensemble RF. All other hyperparameters were held at their default values during each sweep. Performance was highly stable with respect to $n_\\text{estimators}$ and $\\text{max\\_depth}$, while $\\text{max\\_features}$ and larger $\\text{min\\_samples\\_leaf}$ values caused mild degradation. Importantly, the selected configuration was applied consistently across datasets without per-dataset tuning, indicating that RF-Deep does not require extensive hyperparameter optimization during deployment.
>
> | Hyperparameter | Value | Dataset-specific RF | Ensemble RF |
> |---|---|---|---|
> | $n\_\\text{est}$ | 50/1k/2k | 96.3/96.5/96.5 | 94.4/94.4/94.3 |
> | $\\text{max\\_depth}$ | 5/20/None | 96.5/96.3/96.4 | 94.3/94.5/94.5 |
> | $\\text{max\\_feat}$ | $\\sqrt{p}$/$\\log\_2 p$/None | 96.5/96.1/95.4 | 94.4/93.5/94.1 |
> | $\\text{min\\_leaf}$ | 1/4/8 | 96.5/96.2/96.0 | 94.5/94.2/93.8 |
>
> Performance was highly stable across $n_\\text{estimators}$ ($< 0.2$ AUROC variation) and $\\text{max\\_depth}$ ($< 0.2$ AUROC variation). $\\text{max\\_features} = \\sqrt{p}$ was mildly preferred over $\\log_2 p$ and *None*, and increasing $\\text{min\\_samples\\_leaf}$ from 1 to 8 caused slight degradation ($\\sim 0.5$ AUROC). Overall, these ablations support the use of standard RF settings without extensive task-specific tuning.

---

> ### Author Response · Authors · 2026-03-27
> **Response to reviewer (2 of 2)**
>
> **Concern 3: The 10\% no-prediction case**
>
> We have expanded the discussion with quantitative analysis. Across OOD datasets, no-prediction rates varied substantially: RSNA PE (10.1%), MIDRC C19$^-$ (32.3%), KiTS (32.2%), PancreasCT (34.2%). Critically, fewer than 3 out of 98 ID scans produced zero foreground predictions, confirming that the absence of a foreground prediction is itself a reliable OOD signal for populations with confirmed tumors. Additionally, we implemented an enhanced pipeline that auto-flags no-prediction scans as OOD (assigning score $= 1.0$) and applies RF-Deep to the remainder. Results are summarized below.
>
> | Method | RSNA PE | MIDRC C19$^-$ | KiTS | PancreasCT |
> |---|---|---|---|---|
> | *Dataset-Specific* | | | | |
> | *AUROC* | | | | |
> | Standard | 95.80 (92.60--98.50) | 93.30 (89.90--96.00) | 100.00 (99.90--100.00) | 100.00 (100.00--100.00) |
> | Enhanced | 96.14 (92.80--98.29) | 95.43 (92.53--97.34) | 99.42 (99.07--100.00) | 99.44 (99.12--100.00) |
> | *FPR95 (%)* | | | | |
> | Standard | 15.20 (7.10--23.50) | 25.50 (16.80--36.20) | 0.10 (0.00--1.00) | 0.00 (0.00--0.00) |
> | Enhanced | 15.43 (6.61--24.49) | 19.96 (12.73--29.11) | 0.68 (0.00--1.02) | 0.67 (0.00--1.02) |
> | *Ensemble* | | | | |
> | *AUROC* | | | | |
> | Standard | 93.70 (91.70--96.50) | 92.80 (89.80--95.60) | 100.00 (99.80--100.00) | 100.00 (100.00--100.00) |
> | Enhanced | 94.17 (91.39--97.16) | 94.97 (93.18--96.81) | 99.41 (98.92--100.00) | 99.44 (99.12--100.00) |
> | *FPR95 (%)* | | | | |
> | Standard | 19.90 (11.20--29.10) | 32.70 (20.40--54.10) | 0.00 (0.00--0.00) | 0.00 (0.00--0.00) |
> | Enhanced | 20.39 (11.71--29.59) | 26.12 (14.77--41.40) | 0.67 (0.00--1.02) | 0.67 (0.00--1.02) |
>
> The enhanced pipeline introduced only 2.73 mean ID false positives per run (out of 98 ID test scans; approximately 2.8%), indicating that the no-prediction signal does not materially degrade in-distribution specificity. We note that for datasets with low no-prediction rates (RSNA PE: 10.1%), the enhanced pipeline's hard assignment of score $= 1.0$ marginally reduced AUROC relative to RF-Deep alone, since scans that would have been correctly scored by the RF were instead assigned a fixed maximum score. This is expected behavior: the enhanced pipeline is most beneficial when no-prediction rates are high (KiTS, PancreasCT, MIDRC C19$^-$), where it provides complementary anatomically interpretable OOD signal, and least impactful when rates are low. Taken together, these results suggest that absent predictions can be used as a complementary OOD signal with minimal effort and negligible impact on ID specificity. However, as this cannot be completely verified and "red-teamed" with the available datasets for in-distribution, we do not adopt it as the default pipeline.
>
> Of note, this concern was shared by Reviewers QbSa (Concern 3) and uU1P (Concern 5); we provide a unified analysis here and summarize the key results in those responses. This result has been updated in the revised paper (Section 5.2.3).
>
> **Concern 4: Plug-and-play framing**
>
> We agree that "plug-and-play" may overstate the requirements relative to zero-overhead methods like MaxSoftmax. We have revised the manuscript to use better terminology, specifically, "lightweight and post-hoc" throughout and now explicitly enumerate the three requirements for RF-Deep:
>
> 1. A frozen segmentation model (no retraining or architectural modification).
> 2. As few as 40 labeled scans (20 ID + 20 OOD).
> 3. Default RF hyperparameters.
>
> Given our findings in Concern 1, we therefore position RF-Deep as a lightweight, post-hoc detector that offers a favorable accuracy--effort trade-off for risk-aware deployment.
>
> We hope these additions address the reviewer's concerns regarding computational overhead, hyperparameter sensitivity, deployment requirements, and workflow integration. We believe the revised manuscript now presents a clearer and more carefully scoped evaluation of RF-Deep.

---

### Review · Reviewer_QbSa · 2026-03-17

**Summary Of Contributions:**

This work studies scan-level out-of-distribution detection for lung tumor segmentation in 3D CT and proposes RF-Deep, a post-hoc detector that extracts multi-scale encoder features from tumor-anchored ROIs generated by a frozen segmentation model, then trains a random forest with limited outlier exposure. The method is evaluated on one ID dataset and four OOD datasets spanning near-OOD thoracic CT and far-OOD abdominal CT, with comparisons against MaxSoftmax, MaxLogit, energy, radiomics-based RF, and Mahalanobis-distance baselines. The reported results show especially strong gains on the harder near-OOD settings, while also achieving near-perfect performance on far-OOD cases.

**Additional Comments:**

NA

**Audience:**

Yes

**Audience Explanation:**

I think at least part of the TMLR audience would be interested in this paper because it studies a practically important reliability problem in medical image analysis, namely, how to detect out-of-distribution scans for a deployed segmentation system. The method is simple and easy to understand, built on existing models, making the findings relevant not only to medical imaging researchers but also to readers interested in trustworthy machine learning, post hoc uncertainty estimation, and distribution shift. The scope is somewhat specialized because the experiments focus on lung tumor CT segmentation, so the appeal may not be universal across all of TMLR. However, the problem setting and the empirical lessons are still broad enough to be of clear interest to a meaningful subset of the readership.

**Claims And Evidence:**

Yes

**Claims Explanation:**

The main claims are supported by solid and clearly presented experiments. The paper shows consistent improvements over the compared baselines on both near OOD and far OOD datasets, reports standard metrics with confidence intervals, and includes helpful ablations and sensitivity analyses that make the results more convincing. In particular, the evidence strongly supports the central claim that RF Deep works well for scan-level OOD detection in the specific setting of lung tumor CT segmentation. My only reservation is that some broader claims about generalization and plug-and-play use are less fully established, since performance drops in leave one dataset out tests and the method depends on the segmenter producing foreground predictions. Overall, the evidence is convincing for the main practical contribution, but some claims would be stronger if phrased more narrowly.

**Requested Changes:**

- 1.	Please clarify the framing of the method. Right now, the paper sometimes reads as if RF-Deep were a broadly post hoc OOD detector, but in practice, it relies on limited outlier exposure during training. I do not see this as a flaw in itself, but I do think the paper should be more explicit about that setting and avoid overstating how assumption-light the method is.
- 2.	Please tone down the generalization claims. The results are strong in the studied lung tumor CT setting, but the leave one dataset out experiments show that performance on near OOD data drops when the held out OOD type is different from those seen during training. Because of that, I think the paper should present its generalization claims more carefully and more narrowly.
- 3.	Please address the no-foreground case more clearly. Since the method depends on tumor-anchored ROI extraction, an important practical question is what happens when the segmenter predicts no lesion at all. The current manuscript acknowledges this limitation, but I think it needs a clearer explanation of how such cases should be handled in deployment and whether they should themselves be treated as suspicious or OOD.
- 4.	Please narrow the scope of the conclusions. The experiments are all on single-label lung tumor CT segmentation, so I do not think the current evidence is enough to support broader claims about wide applicability across segmentation settings. The conclusions should better match the actual experimental scope.
- 5.	A more direct ablation on tumor anchoring would help. The current comparisons are thoughtful, especially in separating deep features from radiomics and RF from Mahalanobis-style detection, but I still came away wanting a cleaner test of whether tumor-anchored ROIs are truly better than more generic feature extraction strategies.
- 6.	A bit more discussion of baseline coverage would improve the paper. The chosen baselines are reasonable, but readers may still wonder whether stronger recent post hoc OOD baselines, especially in feature space, would change the picture. Even a more explicit discussion would help.

---

> ### Author Response · Authors · 2026-03-18
>
> Thank you for your review and positive feedback. We appreciate your time and thoughtful analysis and will revert back shortly.

---

> ### Author Response · Authors · 2026-03-27
> **Response to reviewer (1 of 3)**
>
> We thank the reviewer for the careful reading, the positive assessment of our contribution, and the constructive suggestions. We address each requested change below with new experiments and manuscript revisions, focusing on stronger feature-space baselines, tumor-anchoring ablations, and expanded external evaluation on blinded validation datasets.
>
> **Concern 1: Clarify the framing of the method**
>
> We have revised the framing throughout the paper and now explicitly describe RF-Deep as a *lightweight, post-hoc* OOD detector that requires limited outlier exposure. As detailed in our response to Reviewer q5h1 (Concern 4), we have removed all uses of "plug-and-play" from the manuscript and now explicitly enumerate the three requirements for deploying RF-Deep:
>
> 1. A frozen segmentation model (no retraining or architectural modification).
> 2. As few as 40 labeled scans (20 ID + 20 OOD).
> 3. Default random forest hyperparameters (no task-specific tuning required).
>
> The abstract, introduction, and discussion have been revised to consistently frame the method within the outlier exposure paradigm, following established terminology. Furthermore, our systematic inference time benchmark (detailed in our response to Reviewer q5h1, Concern 1) and our RF hyperparameter ablation (detailed in our response to Reviewer q5h1, Concern 2) indicate that the approach is lightweight and that the default RF settings are optimal across all tested configurations, reinforcing that RF-Deep only requires the outlier dataset curation itself to function properly. Finally, we have clarified that RF-Deep is not intended as a universal OOD detector but as a targeted post-hoc safety filter for deployment settings where representative failure modes can be anticipated or are recorded.
>
> **Concern 2: Tone down generalization claims**
>
> We agree and have revised the manuscript to carefully frame the scope and generalization claims corresponding to the presented work. The abstract and conclusions now state that RF-Deep achieves strong OOD detection *within the studied lung tumor CT setting*, and the discussion of LODO results explicitly acknowledges that near-OOD performance degrades when the held-out pathology differs from training outliers (RSNA PE: AUROC 91.20 versus 95.80 for dataset-specific; MIDRC C19$^-$: 91.00 versus 93.30). In addition, we note that retrospective multi-center datasets do not allow complete disentanglement of pathology and acquisition effects; our analyses are therefore intended to better understand sensitivity to such confounding rather than eliminate it.
>
> To provide concrete generalization evidence, we evaluated on two blinded validation datasets that were never included in any OE training split (introduced in our response to Reviewer uU1P, Concern 5): COVID-19 positive CT ($N{=}110$, denoted MIDRC C19$^+$), and Breast cancer CT ($N{=}66$). RF-Deep achieved AUROC $\\geq$ 94.63 on both blinded validation datasets across both strategies without any exposure to these categories during training, matching or exceeding performance on the seen near-OOD datasets.
>
> As detailed in our response to Reviewer uU1P (Concern 5), these results held across all training strategies and SHAP feature importance profiles on the blinded validation datasets were structurally indistinguishable from those on the *seen* datasets, suggesting that RF-Deep learns pathology-relevant features rather than dataset-specific signatures. Hence, we frame these results as evidence of generalization *within the thoracic CT domain* rather than claiming universal applicability.
>
> Further, in the revised manuscript, we also consolidated the deployment strategies by removing the Unified and LODO+ configurations. The Unified strategy (a single RF trained on all OOD datasets combined) provided intermediate performance between LODO and Dataset-specific without distinct practical insights beyond what the Ensemble already demonstrates. LODO+ (LODO augmented with non-tumor-centric ROIs) was superseded by the external validation on the blinded validation datasets, which provides a stronger and more direct test of generalization. The streamlined table now focuses on the two operationally distinct deployment modes: Dataset-specific (known failure modes), Ensemble (unknown failure modes), with additional analysis on LODO (held-out evaluation on the original datasets) retained for smoother flow.

---

> ### Author Response · Authors · 2026-03-27
> **Response to reviewer (2 of 3)**
>
> **Concern 3: Address the no-foreground case more clearly**
>
> We have expanded the discussion with quantitative analysis, as also detailed in our responses to Reviewer q5h1 (Concern 3) and Reviewer uU1P (Concern 5). Across OOD datasets, no-prediction rates varied substantially: RSNA PE (10.1%), MIDRC C19$^-$ (32.3%), KiTS (32.2%), PancreasCT (34.2%). Critically, fewer than 3 out of 98 ID scans produced zero foreground predictions ($\\sim$2.8%), confirming that the absence of a foreground prediction is itself a reliable OOD signal for populations with confirmed tumors.
>
> We implemented and evaluated an enhanced pipeline that auto-flagged no-prediction scans as OOD (assigning score $= 1.0$) and applied RF-Deep to the remainder. Results are summarized below:
>
> | Method | RSNA PE | MIDRC C19$^-$ | KiTS | PancreasCT |
> |---|---|---|---|---|
> | *Dataset-Specific* | | | | |
> | *AUROC* | | | | |
> | Standard | 95.80 (92.60--98.50) | 93.30 (89.90--96.00) | 100.00 (99.90--100.00) | 100.00 (100.00--100.00) |
> | Enhanced | 96.14 (92.80--98.29) | 95.43 (92.53--97.34) | 99.42 (99.07--100.00) | 99.44 (99.12--100.00) |
> | *FPR95 (%)* | | | | |
> | Standard | 15.20 (7.10--23.50) | 25.50 (16.80--36.20) | 0.10 (0.00--1.00) | 0.00 (0.00--0.00) |
> | Enhanced | 15.43 (6.61--24.49) | 19.96 (12.73--29.11) | 0.68 (0.00--1.02) | 0.67 (0.00--1.02) |
> | *Ensemble* | | | | |
> | *AUROC* | | | | |
> | Standard | 93.70 (91.70--96.50) | 92.80 (89.80--95.60) | 100.00 (99.80--100.00) | 100.00 (100.00--100.00) |
> | Enhanced | 94.17 (91.39--97.16) | 94.97 (93.18--96.81) | 99.41 (98.92--100.00) | 99.44 (99.12--100.00) |
> | *FPR95 (%)* | | | | |
> | Standard | 19.90 (11.20--29.10) | 32.70 (20.40--54.10) | 0.00 (0.00--0.00) | 0.00 (0.00--0.00) |
> | Enhanced | 20.39 (11.71--29.59) | 26.12 (14.77--41.40) | 0.67 (0.00--1.02) | 0.67 (0.00--1.02) |
>
> The enhanced pipeline maintained strong performance with negligible ID specificity impact. However, as its severity cannot be fully characterized across all deployment scenarios with the currently available in-distribution datasets, we only consider RF-Deep if predictions are present; and flag others for primary manual review. Future work may explore developing a two-stage approach that leverages the no-prediction signal (as well as other surrounding factors) as complementary, anatomically interpretable ID/OOD indicator. This result has been updated in the revised paper (Section 5.2.3).
>
> **Concern 4: Narrow the scope of conclusions**
>
> We have revised the conclusions to match the experimental scope. The revised discussion and conclusion now explicitly state that all evaluations were performed on single-label lung tumor CT segmentation and that generalization to other anatomical sites, multi-label segmentation, and non-CT modalities remains to be established. RF-Deep is framed as a method *demonstrated for* lung tumor CT segmentation, with extension to other settings identified as a direction for future work rather than an implied capability. Further, we are aware that extending RF-Deep to multi-label segmentation or multi-modal applications will likely require additional exposure design and validation.
>
> **Concern 5: Ablation on tumor anchoring**
>
> We conducted a direct ablation isolating the effect of the tumor-anchoring strategy. Using the same RF-Deep pipeline (same backbone features, same RF classifier, same random seeds), we compared three crop strategies:
> - **Anchored (default)**: 4 crops centered on predicted tumor regions (the proposed method).
> - **Spatial**: 4 random crops drawn uniformly from the body volume (no label guidance, matched crop count to anchored).
> - **Center**: A single deterministic crop from the center of the body volume (no label guidance).
>
> The results revealed two complementary findings: (1) the hierarchical encoder features are the primary driver of OOD detection and even without tumor guidance (center and spatial modes), the multi-scale features extracted from the frozen segmentation encoder provide strong OOD discrimination. However, anchoring around the tumor provided consistent additional gains, especially on near-OOD detection. AUROC improved monotonically from center to spatial to anchored on every near-OOD and blinded validation dataset: RSNA PE (+2.10 from center to anchored), breast cancer CT (+2.22), MIDRC C19$^+$ (+2.63). The largest gains appear on the hardest near-OOD cases, where the predicted lesion regions contain the most diagnostically relevant features for distinguishing pathological differences within the same anatomical site. Far-OOD detection is saturated regardless of crop strategy, consistent with the anatomical differences being captured by any crop placement. These results suggest that anchoring primarily stabilizes feature aggregation around task-relevant regions rather than introducing new discriminative signal.
> (contd.)

---

> ### Author Response · Authors · 2026-03-27
> **Response to reviewer (3 of 3)**
>
> (contd.)
> This ablation has been added as Figure 11 in the revised manuscript, using the same format as the existing design-choice ablations. We note that this result also complements the ROI-restricted logit baseline experiments conducted for Reviewer uU1P (Concern 4), which demonstrated that restricting logit-based methods to the same tumor-anchored ROIs used by RF-Deep did not close the performance gap, confirming that the advantage arises from the feature representation rather than spatial focus alone.
>
> Lastly, as our paper mentions, the idea of anchoring is to firmly affirm features for RF-Deep to be around the tumor segmentation. Tumor anchoring also provides clinical interpretability: without it, RF-Deep might extract features from anatomically irrelevant regions (e.g., abdominal structures in a chest pulmonary embolism detected as lung cancer scan), yielding a correct OOD flag for the wrong reasons and obscuring the clinical rationale for the detection.
>
> **Concern 6: Baseline coverage discussion**
>
> We agree that a more explicit discussion of baseline selection is warranted and have added both new experiments and an expanded discussion to the revised manuscript. We selected Mahalanobis distance (MD-Deep) as the primary comparison because it operates on identical backbone features as RF-Deep and is the most established unsupervised feature-space baseline in OOD detection. We expanded this baseline and added experiments with three recent feature-space OOD methods (ReAct [Ref1], ASH [Ref2], X-Mahalanobis [Ref3]) adapted to our 3D segmentation pipeline. All were evaluated on the same set of features and under identical conditions.
>
> ReAct applies percentile-based activation truncation (90th percentile); ASH applies activation sparsification by zeroing values below the 90th percentile. X-Mahalanobis extends standard Mahalanobis distance with adaptive multi-layer feature fusion across transformer stages, directly validating our core design choice that hierarchical features are more informative than final-layer features alone.
>
> | Method | RSNA PE | MIDRC C19$^-$ | KiTS | PancreasCT |
> |---|---|---|---|---|
> | *AUROC* | | | | |
> | MD-Deep | 87.30 (83.90--91.10) | 84.50 (79.40--89.00) | 99.20 (97.90--99.80) | 99.20 (98.20--100.00) |
> | MD + ReAct | 83.25 (78.95--87.41) | 80.45 (74.55--86.63) | 98.68 (96.55--99.62) | 98.27 (95.76--99.67) |
> | MD + ASH | 73.34 (69.23--78.65) | 76.68 (71.70--81.67) | 97.93 (95.80--99.35) | 97.24 (93.90--99.06) |
> | X-Mahalanobis | 85.44 (82.09--88.94) | 81.26 (75.45--87.32) | 98.55 (96.85--99.71) | 98.59 (97.06--99.93) |
> | RF-Deep | 95.80 (92.60--98.50) | 93.30 (89.90--96.00) | 100.00 (99.90--100.00) | 100.00 (100.00--100.00) |
> | *FPR95 (%)* | | | | |
> | MD-Deep | 45.10 (30.60--60.30) | 58.50 (38.20--77.60) | 3.10 (0.00--9.20) | 2.80 (0.00--7.70) |
> | MD + ReAct | 54.62 (40.77--67.88) | 64.76 (43.34--85.23) | 5.79 (1.02--11.76) | 6.39 (1.51--13.27) |
> | MD + ASH | 83.01 (69.34--92.86) | 75.73 (61.71--90.87) | 8.45 (2.04--20.41) | 10.61 (3.55--26.05) |
> | X-Mahalanobis | 51.16 (40.82--63.27) | 63.29 (44.85--83.67) | 3.88 (0.00--9.18) | 4.32 (1.02--12.24) |
> | RF-Deep | 15.20 (7.10--23.50) | 25.50 (16.80--36.20) | 0.10 (0.00--1.00) | 0.00 (0.00--0.00) |
>
> The results showed no improvement over MD-Deep: activation-shaping methods (ReAct, ASH) consistently reduced AUROC, while X-Mahalanobis provided no gains, since our baseline already uses concatenated hierarchical features. Across datasets, an 8--10 AUROC gap remained between the best unsupervised baseline (MD-Deep) and RF-Deep, suggesting that outlier exposure and RF classification, not feature transformations, drive the performance improvement, and further research is required in the non-OE domain. We postulate that activation shaping was counterproductive in this scenario because spatial pooling already dampened the extreme activations and distorted the covariance needed for Mahalanobis distance scoring. This result has been updated in the revised paper (Appendix C). Finally, we also excluded KNN-based approaches and ViM as they are similarly unsupervised, relying solely on ID feature density without outlier exposure that our results show is ineffective for near-OOD detection (this has been acknowledged in Section 4.4 of the revised paper).
>
> [Ref1] Sun, Y., Guo, C. and Li, Y., 2021. React: Out-of-distribution detection with rectified activations. Advances in neural information processing systems, 34, pp.144-157.
>
> [Ref2] Djurisic, A., Bozanic, N., Ashok, A. and Liu, R., 2022. Extremely simple activation shaping for out-of-distribution detection. arXiv preprint arXiv:2209.09858. Accepted at ICLR 2023.
>
> [Ref3] Wei, T., Wang, B.L., Shi, J.X., Li, Y.F. and Zhang, M.L., X-Mahalanobis: Transformer Feature Mixing for Reliable OOD Detection. In The Thirty-ninth Annual Conference on Neural Information Processing Systems. 2025.
>
> We hope these additions in the revised manuscript of RF-Deep address the reviewer's concerns .

---

### Decision · Action_Editor_dk61 · 2026-04-19

**Recommendation:** Accept as is

**Additional Comments:**

The authors have thoroughly addressed all substantive reviewer concerns through additional experiments, clearer framing, and strengthened analysis. The revised manuscript is technically sound, well scoped, and ready for publication

**Audience:**

Yes

**Audience Explanation:**

At least some individuals in TMLR’s audience would be interested in this paper. The work addresses an important problem, namely: OOD detection for deployed deep learning systems, specifically in medical image segmentation, where silent failures can have serious consequences. This aligns well with TMLR interests such as reliability, robustness, post‑hoc safety mechanisms, and trustworthy machine learning.

While the experimental focus is on lung tumor CT segmentation, the broader ideas (i.e., leveraging frozen deep features, limited outlier exposure, and lightweight post‑hoc detectors) are relevant to readers working on: OOD detection and uncertainty estimation, safety filters for high‑stakes ML applications, empirical analysis of robustness under distribution shift, practical deployment of large pretrained models. The method’s simplicity, strong empirical validation, and careful discussion of limitations also make it valuable to practitioners and researchers concerned with real‑world deployment. Although the scope is specialized and may not appeal to the entire TMLR readership, it is well within the interests of a substantial and appropriate subset of the audience.

**Claims And Evidence:**

Yes

**Claims Explanation:**

The claims made in the submission are supported by convincing evidence. The revised manuscript provides a comprehensive empirical evaluation across multiple in‑distribution, near‑OOD, and far‑OOD datasets, with appropriate use of standard metrics (AUROC, FPR95) and confidence intervals. The authors substantiate their central claims (that RF‑Deep improves scan‑level OOD detection for lung tumor CT segmentation) through consistent gains over strong baselines, particularly in challenging near‑OOD settings.

Earlier concerns raised by reviewers (e.g., potential scanner/protocol confounding, fairness of baseline comparisons, reliance on tumor anchoring, and handling of no‑foreground predictions) have been addressed with additional experiments, ablations, and clarifications. The authors also clarified the scope of their conclusions to match the evidence. Overall, the experimental design is careful, the analyses are thorough, and the conclusions are well aligned with the reported results.